# FOXA1 repression drives lineage plasticity and immune heterogeneity in bladder cancers with squamous differentiation

Cancers arising from the bladder urothelium often exhibit lineage plasticity with regions of urothelial carcinoma adjacent to or admixed with regions of divergent histomorphology, most commonly squamous differentiation. To define the biologic basis for and clinical significance of this morphologic heterogeneity, here we perform integrated genomic analyses of mixed histology bladder cancers with separable regions of urothelial and squamous differentiation. We find that squamous differentiation is a marker of intratumoral genomic and immunologic heterogeneity in patients with bladder cancer and a biomarker of intrinsic immunotherapy resistance. Phylogenetic analysis confirms that in all cases the urothelial and squamous regions are derived from a common shared precursor. Despite the presence of marked genomic heterogeneity between co-existent urothelial and squamous differentiated regions, no recurrent genomic alteration exclusive to the urothelial or squamous morphologies is identified. Rather, lineage plasticity in bladder cancers with squamous differentiation is associated with loss of expression of *FOXA1*, *GATA3*, and *PPARG*, transcription factors critical for maintenance of urothelial cell identity. Of clinical significance, lineage plasticity and PD-L1 expression is coordinately dysregulated via *FOXA1*, with patients exhibiting morphologic heterogeneity pre-treatment significantly less likely to respond to immune checkpoint inhibitors.

For decades, platinum-based combination chemotherapy was the primary systemic treatment for patients with metastatic bladder cancer[1–3]. More recently, immune checkpoint inhibitors were shown to induce profound and durable responses in patients with metastatic bladder cancer who progressed after platinum-based chemotherapy[1,3–6]. However, a majority of bladder cancers are intrinsically resistant to immune checkpoint inhibitors and even for those patients that respond, acquired resistance is a major clinical challenge.

Cancers arising from the bladder urothelium display a wide spectrum of histomorphologies, with regions of urothelial carcinoma often adjacent to or admixed with regions of squamous, glandular, sarcomatoid, small cell, or other histologic variants[7–10]. These morphologic variants are often underreported and unrecognized[9,10]. While

genomic differences have been reported between morphologically distinct regions of some mixed histology tumors[11,12], the molecular basis for morphologic heterogeneity in bladder cancer has not been systematically studied.

Transcriptomic-based molecular classification of urothelial carcinomas has revealed that the basal subtype is enriched for tumors with squamous differentiation, the most common bladder cancer variant[13,14]. As bladder tumors of the basal subtype are less likely to respond to the anti-PD-L1 antibody atezolizumab[4], we hypothesized that bladder cancers with mixed histology and regions of squamous differentiation may be intrinsically resistant to immunotherapy due to the presence of a pre-existent treatment-resistant population of cancer cells. To date, biomarker analyses incorporated into immunotherapy

✉e-mail: ddegraff@pennstatehealth.psu.edu; alahmadh@mskcc.org

clinical trials have been unable to address this question as only a single tumor region is typically analyzed by bulk sequencing. While such analyses can yield important insights, they cannot account for the confounding effects of intra-tumoral and tumor-to-tumor heterogeneity. To determine whether morphologic heterogeneity is a marker for genomic and immunologic heterogeneity in bladder cancers with mixed histology, we performed an integrated genomic analysis of morphologically and spatially distinct regions of urothelial and squamous morphology from tumors with heterogeneous histopathology.

Here, we show that the urothelial and squamous regions are derived from a common shared precursor but there is no recurrent genomic alteration exclusive to the urothelial or squamous morphologies. We also show that lineage plasticity in urothelial carcinoma with squamous differentiation is associated with loss of expression of transcription factors critical for maintaining urothelial cell identity and luminal phenotype such as *FOXA1*, *GATA3*, and *PPARG*. By functional analysis, we show that lineage plasticity and PD-L1 expression is coordinately dysregulated via *FOXA1* repression. Morphologic heterogeneity in the form of squamous differentiation is a marker of intratumoral genomic and immunologic heterogeneity in patients with bladder cancer and a biomarker of intrinsic immunotherapy resistance. Of clinical significance, patients whose tumors exhibit morphologic heterogeneity pre-treatment are significantly less likely to respond to immune checkpoint inhibitors.

## Results

### Regions of urothelial and squamous differentiation have a shared clonal origin

To determine whether there were differences in the genomic landscape of bladder cancers with urothelial and squamous differentiation, we performed a central histologic review of a prospectively sequenced cohort of 848 primary bladder tumors. We found that 69% (587/848) had a pure urothelial carcinoma histology (UC), whereas 82 (10%) had morphologic evidence of squamous differentiation (SqD). Targeted sequencing of 341 or more cancer-associated genes (MSK-IMPACT), revealed higher rates of *TP53* and *CDKN2A* alterations in SqD vs UC but no somatic alteration exclusive to the UC or SqD tumors (Supplementary Fig. 1). We next performed whole exome sequencing (WES) of paired, macrodissected regions of UC and SqD of 21 bladder tumors in which distinct and separable UC and SqD regions were present (total 42 whole exomes) (Fig. 1a). Mean tumor mutational burden (TMB) was higher in the SqD (10.2 mutations/megabase [mut/Mb]) versus the UC regions (7.8 mut/Mb, $p = 0.02$, Wilcoxon paired test, Fig. 1b). Consistent with the higher TMB, the SqD regions also had a higher neoantigen burden than the patient-matched UC regions ($p < 0.001$, Wilcoxon paired test, Supplementary Fig. 2a). The SqD regions also exhibited higher karyotypic complexity with a higher median ploidy relative to the paired UC regions (median 2.6 vs 2.3, $p = 0.02$, Wilcoxon paired test, Supplementary Fig. 2b). Consistent with the genomic landscape of UC as previously defined by the Cancer Genome Atlas (TCGA) and others[13,15], known or presumed oncogenic mutations were identified in *TP53*, chromatin modifying genes (*KMT2D, ARID1A, KMT2C, KDM6A*), and mitogenic signaling pathways (*PIK3CA, FGFR3, ERBB2*) in both UC and SqD regions (Fig. 1c).

Phylogenetic analysis of the WES data confirmed that the morphologically distinct regions of all 21 tumors arose from a shared precursor with a median of 68 (range 16 – 852) non-synonymous mutations shared between the UC and SqD regions (Fig. 1d, e, Supplementary Fig. 3). Clonality analysis further supported that the two components were clonally related and did not arise independently ($p < 0.001$, Supplementary Table 2)[16,17]. To better quantitate the relative burden of private versus shared mutations in the SqD and UC regions, we calculated a "phylogenic ratio" for each pair, defined as the number of private mutations divided by the number of shared mutations. The median phylogenic ratio was 1.1 in the SqD and 0.8 in UC

regions, indicating that there was a similar number of private mutations in each of the morphologically distinct regions. However, in six tumors (30%), the phylogenic ratio of the SqD region was over 4-fold higher than the patients' matched UC sample, suggesting that the SqD component in these cases had evolved significantly further than the UC component from the inferred common shared precursor (Cases 2, 3, 9, 10, 11, and 12). In sum, the data were indicative of early-branched evolution.

While there was significant mutational discordance between the UC and SqD regions of individual tumors, mutational concordance was higher for known or presumed oncogenic mutations (63.2% of oncogenic/likely oncogenic mutations were shared versus 33.9% of all nonsynonymous mutations) with some notable discordant mutations (Fig. 1f). For example, *FGFR3* S249C, an oncogenic and therapeutically actionable hotspot mutation, was detected in only the SqD regions of two cases (tumors 2 and 12, Fig. 1d, e and Supplementary Fig. 3). WES did not, however, identify a specific gene or pathway that was significantly more frequently mutated in the SqD or UC regions. Additionally, tumors from UC and SqD regions had a high degree of concordance in their mutational signatures, with an APOBEC mutational signature the most common mutational signature in most of the paired regions[18] (Supplementary Fig. 4).

### Lineage plasticity in bladder cancers with squamous differentiation

As targeted and whole exome DNA sequencing did not identify a shared mutational alteration or genomic signature unique to the SqD or UC components, we next sought to determine whether dysregulation of gene expression was the basis for the lineage plasticity observed in bladder cancers with mixed UC-SqD histology. We, therefore, performed expression profiling of separable SqD and UC regions of 12 bladder tumors from the WES cohort for which RNA quality was sufficient for whole transcriptome RNA-sequencing (RNAseq, total of 24 regions). Based on centroid clustering analysis of expression data, all 12 SqD regions were basal-squamous subtype based on the TCGA molecular classification schema[13]. Notably, 8 of the UC regions were also basal-squamous subtype, with only four pairs exhibiting discordant transcriptional subtypes (three UC regions were luminal-infiltrated, one luminal-papillary, Fig. 2a).

To better quantify differences in the transcriptional profiles between the paired UC and SqD regions, we performed single sample gene set enrichment analysis (ssGSEA)[19,20]. This revealed that the SqD regions expressed higher levels of basal-squamous genes based on the BASE47 gene set[21] ($p = 0.07$, paired Wilcoxon rank sum test) and UPK/KRT gene sets[20] ($p < 0.001$, paired Wilcoxon rank sum test), whereas the matched UC regions expressed higher levels of luminal genes based on the BASE47 ($p = 0.06$, paired Wilcoxon rank sum test) and UPK/KRT gene sets ($p = 0.02$, paired Wilcoxon rank sum test) (Fig. 2b). These differences were observed even in cases in which both the UC and SqD regions were basal-squamous subtype (Fig. 2b).

To determine if quantitative differences in the expression of basal-squamous genes were identifiable at the single cell level in tumors with squamous morphology, we performed single cell RNA sequencing (scRNAseq) of a bladder urothelial carcinoma with extensive squamous differentiation. After excluding immune and stromal cell populations (Supplementary Fig. 5), we identified five distinct clusters of tumor cells (labeled as 1–5, Fig. 2c, Supplementary Figs. 6a, b). Trajectory inference analysis indicated a linear progression of these populations that phenotypically mirror the differentiation pattern of normal epidermis (Fig. 2d). Cells in cluster 5 were phenotypically similar to basal cells of the urothelium as well as skin epidermis with high expression of Keratin 5 (*KRT5*) and ribosome protein genes suggesting a high proliferative index (Fig. 2e). In contrast, cells in clusters 3 and 4 were phenotypically similar to early differentiation suprabasal (spinous) squamous cells with high levels of desmocollins and desmogleins

(*DSC2/DSG3*) and transglutaminase (*TGM1*). Finally, clusters 1 and 2 resembled late differentiation (granular) squamous cells, with over-expression of filaggrin (*FLG*), as well as activated keratinocytes in an intermediate state of cell differentiation between basal and suprabasal cells, with overexpression of type I cytoskeletal keratins *KRT16* and *KRT13*[22–25] (Fig. 2e, Supplementary Fig. 6a). Notably, many of the genes that were up-regulated by bulk RNA-seq analysis in the SqD regions of tumors with mixed histology were also up-regulated in sub-population 2 of this squamous differentiated tumor (Supplementary Figs. 5c and 6c). Finally, despite the extensive squamous morphology evident in this tumor, some cells within clusters 1–3 had relatively high expression of *FOXA1* (Fig. 2e), suggesting that a subset of tumor cells continued to express luminal differentiation markers. In sum, our bulk and sc-RNAseq results suggest that SqD in bladder tumors with mixed histology can be viewed as a continuous/gradual process that recapi-

tulates the differentiation stages of the normal epidermis, with the squamous morphology representing the extreme of the basal-squamous phenotype. The bulk and single-cell transcriptomic analyses also provide molecular evidence that such a transition is well underway in regions of UC prior to discernable morphologic evidence of squamous differentiation at the histologic level, highlighting the limitations of classifying tumors based on morphologic assessments under light microscopy.

## Regions of SqD in urothelial carcinomas with mixed histology are characterized by loss of FOXA1, GATA3, and PPARγ expression

To identify individual genes contributing to the squamous phenotype, we further analyzed the bulk RNAseq data to identify genes differentially expressed in the UC or SqD regions of mixed

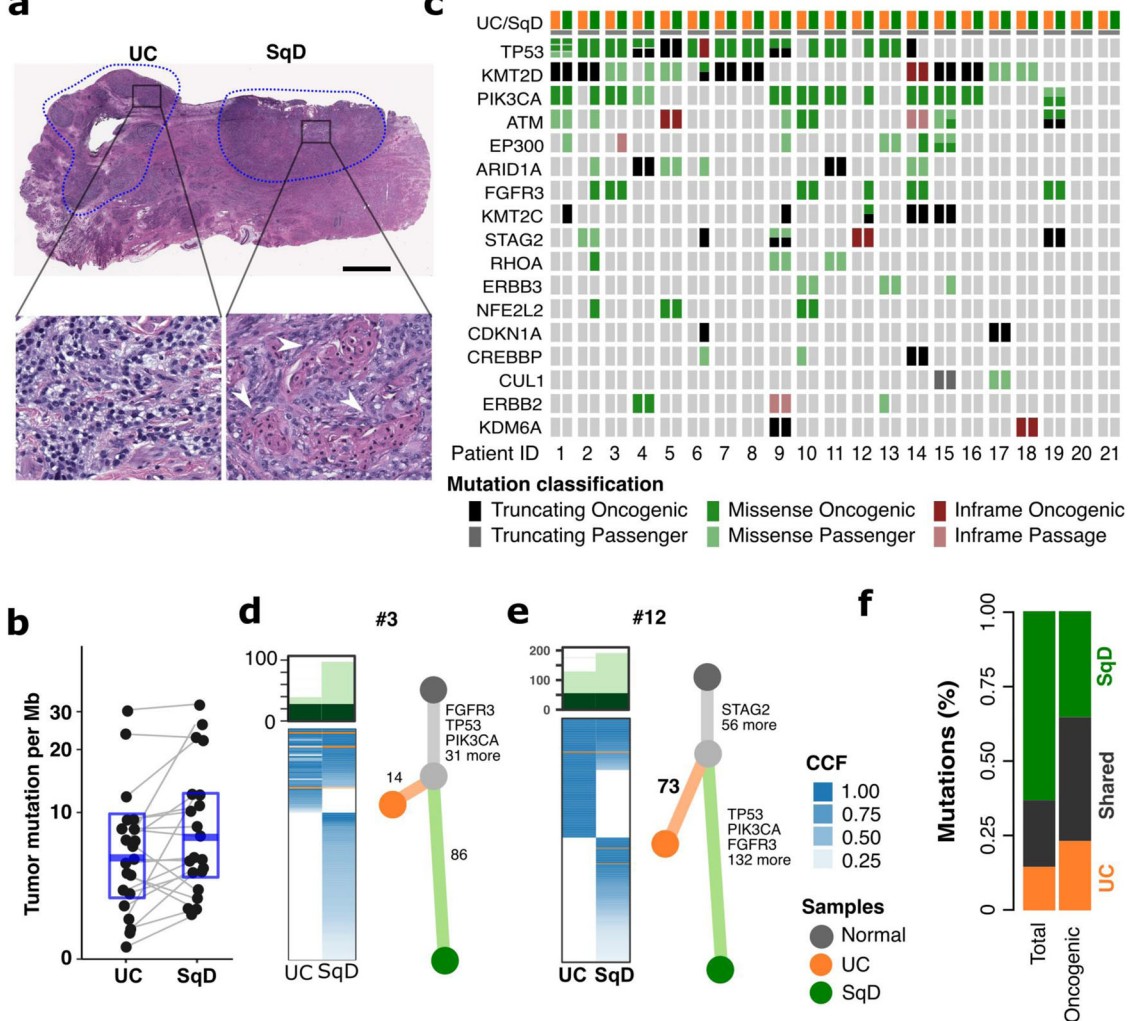

**Fig. 1 | Genomic analysis of urothelial carcinomas with squamous cell differentiation. a** Hematoxylin and Eosin (H&E) staining of a representative mixed histology bladder cancer with discrete and separable regions of urothelial carcinoma, NOS (UC), and squamous differentiation (SqD). In comparison to the urothelial component, the squamous regions were characterized by the presence of tumor cells with keratin formation (block arrow), a microscopic feature unequivocal for squamous differentiation. Scale bar = 2 mm. **b** Whole exome sequencing of macrodissected UC and SqD regions of 21 mixed histology bladder cancers revealed a significantly higher tumor mutational burden in the SqD versus UC regions (two-sided Wilcoxon paired test, *p* 0.02). **c** Oncoprint showing the mutation status of genes commonly mutated in bladder cancer. For each patient, the mutation status

in the UC component is shown on the left, and the SqD component on the right. **d, e** Phylogenetic analysis of two representative urothelial carcinomas with squamous differentiation highlighting that the UC and SqD components were derived from a shared precursor. The number of non-synonymous mutations in each region is shown on top with shared mutations represented in dark green and mutations private to each component in light green. The cancer cell fraction (CCF) of individual mutations is indicated by the degree of blue shading. The UC (orange), SqD (green), and their hypothetical normal cell of origin (gray) are indicated by the different colors. **f** Average of concordant and discordant mutations between paired UC and SqD samples revealed higher concordance of oncogenic and likely oncogenic mutations (63.2%) versus all nonsynonymous mutations (33.9%).

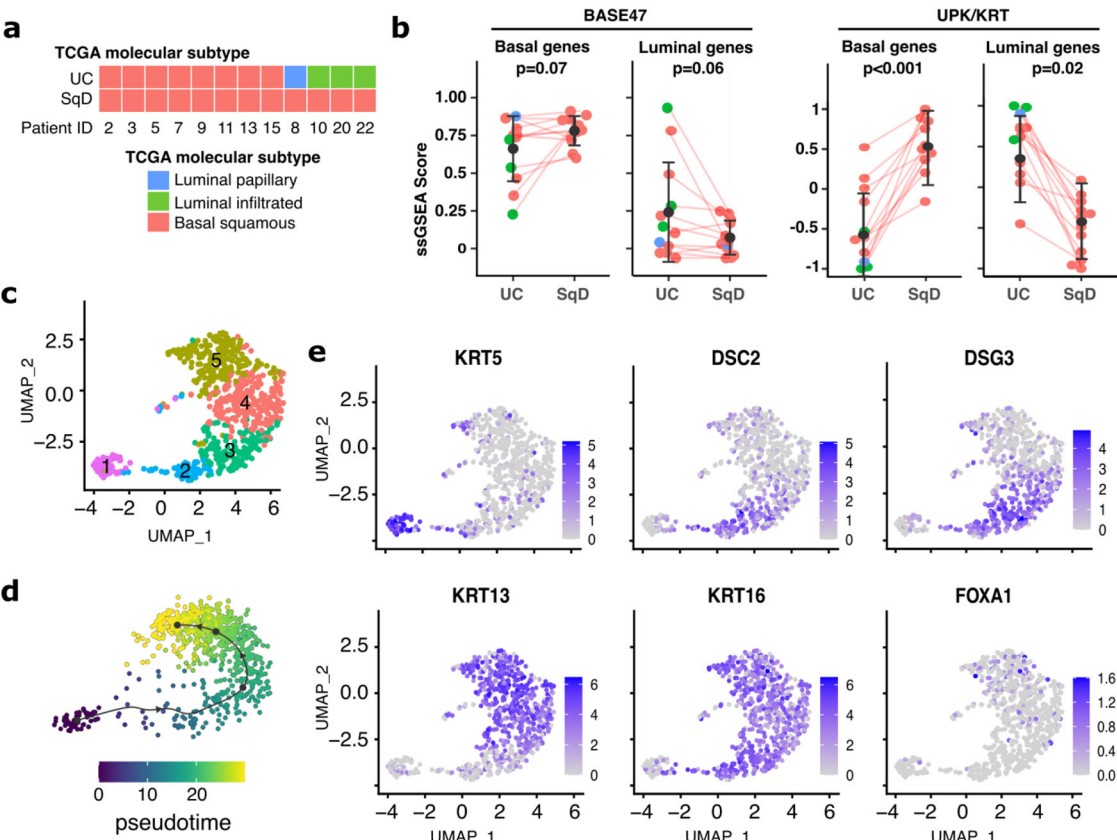

**Fig. 2 | SqD and UC regions of mixed histology tumors have distinct gene expression profiles. a** RNA sequencing analysis of 12 pairs of macrodissected UC and SqD regions from bladder tumors with mixed histology. Each region was classified using the TCGA molecular signature classification schema. All 12 SqD regions were basal/squamous subtype as were 8 of the 12 UC regions. Four tumors had discordant molecular subtypes. **b** Expression of BASE47 (left) and UPK/KRT (right) genes were evaluated by single sample GSEA analysis. Analysis was performed on 12 biologically independent UC-SqD pairs, Error bars represent standard error, *p* = 0.0005, two-sided paired Wilcoxon rank sum test. **c** Single-cell RNA-Seq analysis of a bladder carcinoma with extensive squamous differentiation revealed five clusters of tumor cells (clusters 1–5). **d** Trajectory analysis of these cell

populations indicated a linear progression from cells resembling basal cells to those more similar to differentiated squamous cells. **e** Single-cell expression of select genes. Cluster 1 (61 cells) had an expression signature characteristic of the basal phenotype (high KRT5). Cluster 2 (64 cells) and Cluster 3 (152 cells) had expression signatures indicative of early squamous differentiation similar to that of suprabasal (spinous) cells (stratum spinosum), which are normally located immediately superficial to the basal cells of the epidermis (high DSC2, DSG2). Clusters 4 (231 cells) and 5 (220 cells) had a late squamous differentiation signature similar to that of granular squamous cells (stratum granulosum), which are normally located between the stratum spinosum and stratum corneum of the epidermis (high KRT13, KRT16).

histology tumors. We identified 718 significantly upregulated and 651 downregulated genes in the SqD as compared to the UC regions (Fig. 3a; Fold Change >2 and FDR < 0.05). A subset of the genes upregulated in the SqD regions were genes previously reported to be highly expressed in squamous carcinomas of various lineages, including the peptidase kallikrein genes *KLK13*[26], *KLK10*[27], and *KLK12*[28], the cornification-related genes *CNFN* and *PLA2G4E*[29], the small proline-rich protein genes *SPRR2D* and *SPRR2E*[30], the keratin genes *KRT16*[31], *KRT31*[32] and *KRTDAP*[33], the interleukin *IL36G*[34], and the desmogleins *DSG1/3/4* and desmocollins *DSC2/3*[35,36] (Fig. 3b). Consistent with the sc-RNAseq data above, GSEA also revealed that the top two gene sets upregulated in the SqD regions included genes known to play a role in normal epidermis development and genes associated with the cornified envelope (Fig. 3c, Supplementary Figs. 7a–c). Conversely, genes significantly upregulated in the UC regions compared to SqD regions included *FOXA1*, *GATA3*, and *PPARG*, transcription factors that play a critical role in maintenance of urothelial cell identity[20,37–39] (Wilcoxon signed-rank test, Supplementary Fig. 7d). Immunohistochemistry further confirmed the significantly decreased expression of FOXA1, GATA3 and PPARγ in the SqD versus UC regions of mixed histology bladder tumors (Fig. 3d,

Supplementary Table 1, all *p* < 0.05, paired Wilcoxon test, H-score)[40].

## Intratumoral morphologic heterogeneity is associated with intratumoral immunologic heterogeneity

Intratumoral genetic heterogeneity can result in a large subclonal mutational burden that may facilitate the selection of tumor subclones resistant to immune checkpoint blockade[41]. We, therefore, hypothesized that morphologic heterogeneity could be a predictive biomarker of poor response to anti-PD-1/PDL-1 therapies in bladder cancer patients. To test this hypothesis, we performed a detailed histologic review of bladder tumors collected from 29 patients with locally advanced or metastatic bladder cancer that were treated with the anti-PD-L1 antibody atezolizumab on a therapeutic protocol (clinicaltrials.gov NCT02108652)[6]. This analysis was restricted to performing histopathologic review of hematoxylin and eosin-stained slides from the tumors included in this study and correlation with response to atezolizumab therapy and did not include use of any genomic data from that study[6]. Of the 9 patients who experienced durable clinical benefit, histopathologic evaluation revealed overwhelming tumor morphologic monotony and a lack of morphologic heterogeneity in 7 patients (78%). In contrast, 16 of 20 patients (80%) who derived no

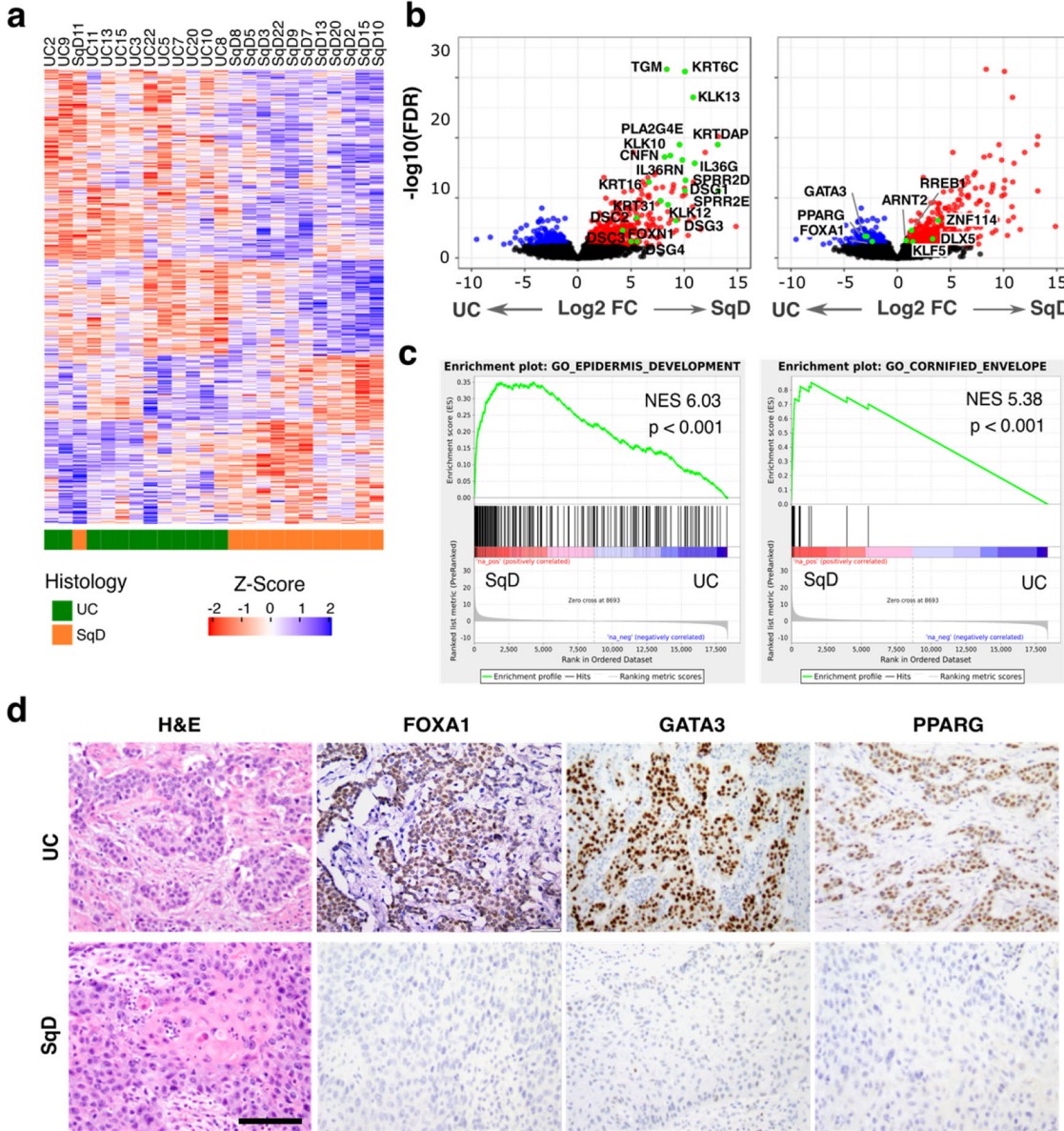

**Fig. 3 | SqD regions are characterized by loss of expression of the FOXA1, GATA3, and PPARG transcription factors. a** Differential gene expression analysis of RNAseq data from 12 biologically independent samples following macrodissection of paired UC and SqD regions revealed 718 significantly upregulated and 651 significantly downregulated genes in the SqD versus the UC regions (defined as log2 FoldChange >1 and FDR < 0.05). **b** Volcano plot of the same 12 UC-SqD pairs showing differentially expressed genes with log FoldChange on the x-axis and log *p*-value on the y-axis. Highlighted are genes associated with urothelial differentiation (left, blue color) or squamous differentiation during normal skin development (right, red color). **c** Gene Set Enrichment Analysis (GSEA) identified gene sets associated with epidermis development and the cornified envelope as the top gene sets upregulated in the SqD versus UC regions (two-sided paired Wilcoxon test). **d** Representative H&E and immunohistochemical stains showing loss of FOXA1, GATA3, and PPARγ protein expression exclusive to the SqD regions of mixed histology tumors. Scale bar = 500 μm.

clinical benefit from atezolizumab had tumors with mixed histology (*p* = 0.01, Fisher's exact test, Fig. 4a).

Given the association between heterogeneous histomorphology and intrinsic resistance to immune checkpoint blockade, we investigated whether variations in tumor histopathology were associated with heterogeneity of the immune tumor microenvironment. More specifically, we performed tumor immune subtype analysis[42] and immune cell deconvolution analysis of the RNAseq data of the macrodissected UC and SqD regions analyzed above. A C2-interferon-gamma (IFNγ)-dominant immune signature was observed in all 12 SqD regions and 9 UC regions. This immune signature is characterized by significant enrichment for M1/M2 macrophage and CD8 expressing T-cells, and high T-cell receptor diversity[42]. Discordant immune

signatures were noted in the paired UC-SqD regions of three tumors (two UC regions were classified as C1-wound healing and one as C3-inflammatory subtype) (Fig. 4b). Immune cell deconvolution analysis of the macrodissected UC and SqD regions further revealed that the UC regions were enriched for signatures of M2 macrophages (*p* = 0.012), CD56 bright (active) NK cells (*p* < 0.001), and cytotoxic (CD8+) and effector memory T (T$_{EM}$) cells (*p* = 0.03) (for all analyses, paired sample Wilcoxon test was used), whereas SqD regions were enriched for M0 macrophages (Fig. 4c, d, Supplementary Fig. 8). In sum, the results suggest that there is a predisposition in SqD tumors for an IFNγ dominant immune signature and that there are spatial differences in the immune cell microenvironment within tumors of mixed histology.

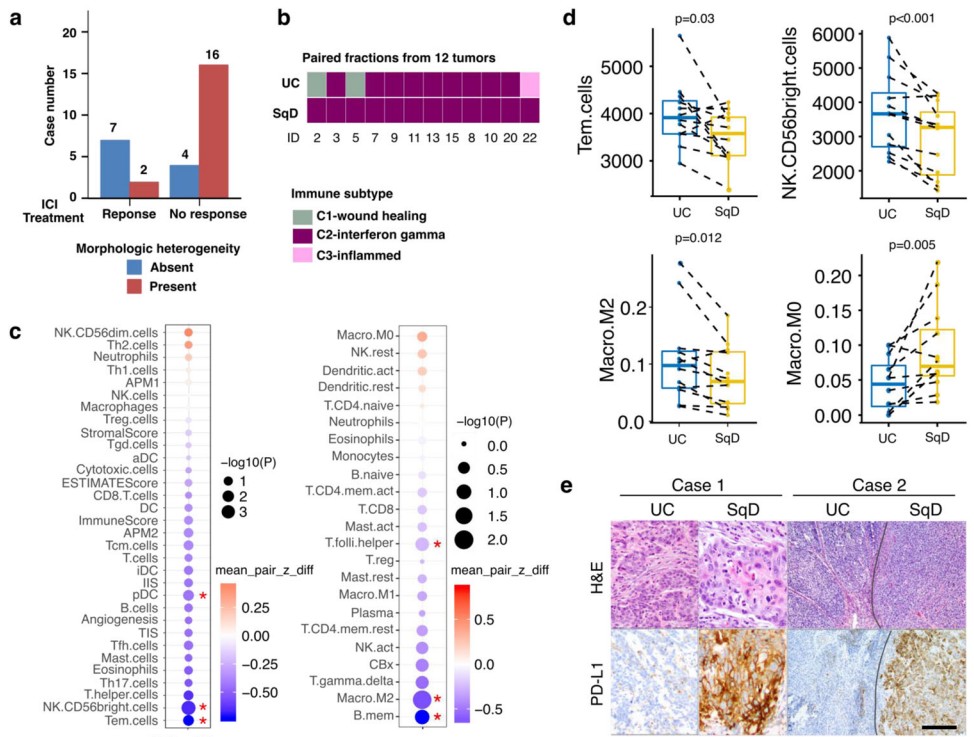

**Fig. 4 | Distinct immune response gene signatures in paired UC and SqD samples. a** Morphologic heterogeneity was associated with a lack of clinical benefit in bladder cancer patients treated with the anti-PD-L1 antibody atezolizumab ($n = 29$) ($p = 0.01$, Fisher's exact test). **b** RNAseq data was used to classify the UC and SqD components from the 12 UC-SqD pairs based on TCGA immune subtypes. **c** Immune cell deconvolution analysis was performed using single sample Gene Set Enrichment analysis (ssGSEA, left) or the CIBERSORTX algorithm (right). The size of the circles is reflective of the *p*-value and the red * indicates significantly enriched immune cells in either the SqD or UC fractions (two-sided Paired sample Wilcoxon test). **d** Immune cell deconvolution scores for individual UC-SqD paired regions were plotted for significantly enriched immune cell types identified by either

ssGSEA or CIBERSORTX. Analyses were performed on 12 biologically independent samples following macrodissection of paired UC and SqD regions. All analyses used paired sample Wilcoxon test. The line in the center of box plot denotes the median. The lower and upper bounds of the box indicate 1st and 3rd quartiles, respectively. The whisker reaches to the maximum and minimum point within the 1.5x interquartile range. Data beyond the end of the whiskers are outliers. **e** PD-L1 immunohistochemical analysis (clone SP263) of two representative tumors showing significantly higher PD-L1 expression in the SqD region versus the adjacent UC region ($p = 0.02$, Wilcoxon rank sum test, also Supplementary Table 1) of mixed histology tumors. Scale bar = 1 mm. Additional microscopic images from Case 2 taken at higher magnifications are included in Supplementary Fig. 13.

We hypothesized that the immune exclusion signature of SqD as compared to UC regions may partly explain the low response rate to immune checkpoint inhibitors in basal-squamous UC tumors despite the dominant IFNγ immune signature[4,43,44]. Factors that can contribute to spatial differences in immune cell populations include (1) regional differences in the expression of cellular proteins that play a mechanical barrier function such as filaggrin (*FLG*) and desmosomal proteins such as *DSC3*[45], and (2) metabolic barriers related to increased hypoxia, decreased fatty acid metabolism and increased glycolysis[46,47], all of which were preferentially altered in SqD regions as compared to UC regions in our cohort (Supplementary Fig. 8).

To validate the difference in immune cell infiltrates between UC and SqD regions noted by immune cell deconvolution analysis of the RNAseq data, we performed immunohistochemical staining for PD-L1 and observed significantly higher PD-L1 protein expression on tumor cells in the SqD as compared to UC regions of 14 of 15 tumors (Fig. 4e and Supplementary Table 1, $p = 0.02$, paired Wilcoxon rank sum test). The tumor-infiltrating immune cells in SqD regions also had greater PD-L1 expression than the tumor-infiltrating immune cells in the UC regions ($p = 0.02$, paired Wilcoxon rank sum test). As illustrated by the two examples shown in Fig. 4e, the SqD regions had strong and diffuse PD-L1 expression whereas the UC regions were largely PD-L1 negative. Multiplex immunofluorescence staining confirmed the high PD-L1 expression on the tumor cells, particularly those with squamous features (Supplementary Fig. 9). In sum, our analyses indicate that morphologic heterogeneity is a marker for both genomic and

immunologic intratumoral heterogeneity and that patients with mixed histology tumors have heterogeneous PD-L1 expression and are less likely to derive clinical benefit from anti-PD-L1 therapy.

## FOXA1 is a bladder cancer cell-intrinsic repressor of the IFNγ transcriptional signature and *CD274*/PD-L1 expression

FOXA1 has previously been shown to regulate PD-L1 (*CD274*) expression in regulatory T-cells[48]. As the squamous regions of mixed histology tumors had both upregulation of PD-L1 and downregulation of FOXA1 expression, we hypothesized that urothelial to squamous lineage plasticity and immune heterogeneity were coordinately dysregulated in mixed histology bladder cancers through changes in *FOXA1* gene expression. To explore this possibility, we generated a series of *FOXA1* knockout (FOXA1-KO) sublines through CRISPR-Cas9 mediated gene editing of UM-UC-1 bladder cancer cells. Loss of FOXA1 expression, as shown by quantitative PCR and immunoblot analysis, was sufficient to induce PD-L1 mRNA and protein expression in multiple UM-UC-1 single-cell clones (Fig. 5a, Supplementary Figs. 10a, b). Conversely, ectopic expression of FOXA1 in UM-UC-3 cells, a bladder cancer cell line with high basal PD-L1 expression, was sufficient to downregulate CD274/PD-L1 expression (Fig. 5b, Supplementary Fig. 10c, d). To characterize the global transcriptional changes resulting from FOXA1 loss-of-function in bladder cancer cells, we performed whole transcriptome RNA sequencing of the parental UM-

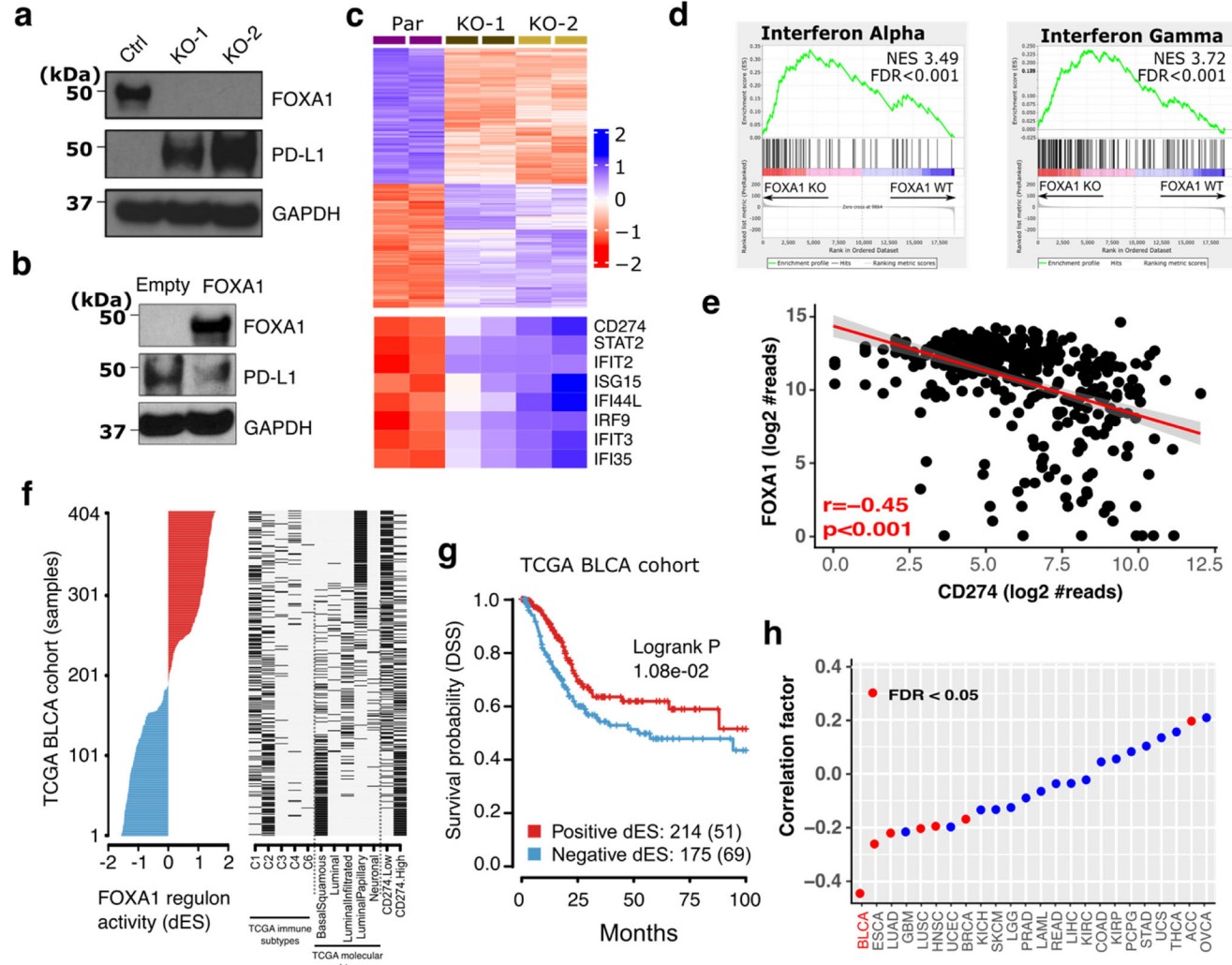

**Fig. 5 | Genetic ablation of *FOXA1* in bladder cancer cells results in increased expression of PD-L1 and interferon sensitive genes (ISGs). a** CRISPR-Cas9 gene editing was used to generate isogenic UM-UC-1 bladder cancer cells with loss of FOXA1 expression. Immunoblot analysis showing that loss of FOXA1 expression was associated with increased PD-L1 expression. This analysis was performed in triplicate. Source data are provided as a Source Data file. **b** UM-UC-3 cells were transfected with FOXA1 or with empty vector. Immunoblot analysis was then performed to quantitate expression of FOXA1, PD-L1, and GAPDH as a loading control. This analysis was performed in triplicate. **c** Differential gene expression (FDR *q* < 0.05) in parental and FOXA1-KO UM-UC-1 bladder cancer cells. Loss of FOXA1 expression was associated with upregulation of 1358 genes including CD274 (PD-L1) and other interferon response genes, and downregulation of 2187 genes. **d** Gene set enrichment analysis (GSEA) identified the interferon alpha and gamma pathways as the

top two gene sets altered following FOXA1 knockout in UM-UC-1 bladder cancer cells. **e** *FOXA1* mRNA expression was significantly negatively correlated with *CD274* (which encodes PD-L1) mRNA expression in the TCGA bladder cancer cohort (two-sided Spearman correlation). **f** FOXA1 regulon activity profiles for 404 samples in the TCGA BLCA cohort sorted by FOXA1 regulon activity (left), molecular subtypes and CD274 gene expression (right). **g** Kaplan-Meier plot of the bladder TCGA cohort[13] showed that disease-specific survival (DSS) was significantly associated with positive vs. negative FOXA1 regulon activity status (Log-rank *p*-value = 0.01). Numbers indicate patients in each group and, in curved parentheses, deceased patients. **h** Correlation between *FOXA1* and *CD274* mRNA expression across 24 cancer types from the TCGA analysis showed that the most significant inverse association between these genes was observed in bladder cancer.

UC-1 cells and two FOXA1-KO sublines. Differential gene expression analysis identified 1358 significantly upregulated and 2187 significantly downregulated genes in the FOXA1-KO as compared to the parental UM-UC-1 cells, including many interferon target genes such as *CD274 (PD-L1)*, *ISG15*, and *IFI44L* (Fig. 5c, FDR < 0.05). Furthermore, analysis of the upregulated genes via GSEA suggested a critical role for FOXA1 in repression of interferon target genes (Fig. 5d, and Supplementary Fig. 10e). We also confirmed that there was a reciprocal relationship between FOXA1 and PD-L1 expression in human bladder cancers using RNA sequencing data from the bladder cancer TCGA cohort (Fig. 5e; Spearman correlation; r = −0.45; *p* < 0.001). This finding was further supported by analyzing FOXA1 regulon activity in paired UC and SqD regions. Regulon refers to target gene sets

controlled (induced and/or repressed) by the same regulator genes[13], and in our cohort, FOXA1 regulon activity was decreased in SqD compared to UC regions of individual bladder cancers (paired Wilcoxon rank sum test, Supplementary Fig. 11). Finally, to further validate our findings, we re-analyzed FOXA1 regulon activity across molecular subtypes in the TCGA bladder cancer cohort[13] and in another previously published study[49]. This analysis confirmed the presence of decreased FOXA1 regulon activity in basal/squamous tumors as compared to luminal tumors and identified a significant inverse correlation between FOXA1 regulon activity and PD-L1 expression in both cohorts (Fig. 5f, Supplementary Fig. 11). Decreased FOXA1 regulon activity was also associated with shorter overall and disease-specific survival in the bladder TCGA cohort (Logrank *p* = 0.01, Fig. 5g, Supplementary

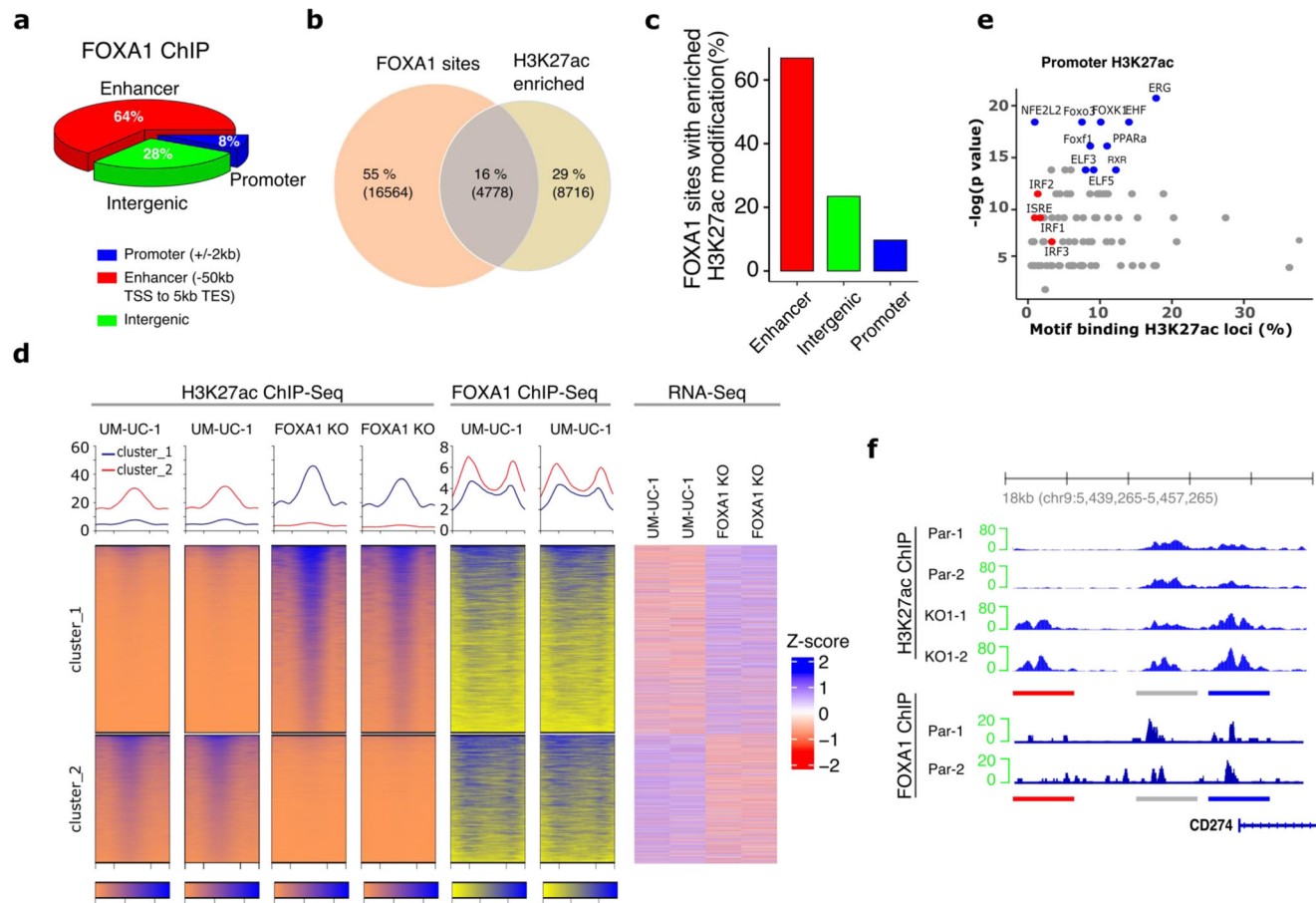

**Fig. 6 | FOXA1 knockout was associated with widespread enhancer and promoter epigenetic reprogramming in bladder cancer cells. a** ChIP-Seq was performed in duplicates using chromatin extracted from UM-UC-1 parental and UM-UC-1 FOXA-1-KO cells. 21,766 FOXA1 binding sites were identified in the ChIP-Seq data by MACS2 (FDR < 0.001). Genomic regions where FOXA-1 binding sites were located were then annotated as promoter (2 kb+/− transcriptional start site−TSS), enhancer (between −50 kb from TSS and 5 kb after transcriptional end site−TES), or intergenic (other sites). **b** Integrated ChIP-Seq analysis of FOXA1 binding sites and H3K27ac revealed 14971 differentially modified H3K27ac loci enriched in either parental UM-UC-1 cells or those with FOXA1 KO (55% were unique FOXA1 binding, 29% were histone H3K27ac enriched sites, and 16% overlapping sites, defined as within 5 kb), **c** The majority of the overlapping sites were in enhancer regions with fewer sites in the intergenic or promoter regions. **d** Cluster analysis of the H3K27ac-modified enhancer regions enriched in either the parental UM-UC-1 cells or the isogenic FOXA1-KO cells (left), FOXA1 binding coverage around the same region (middle), and the expression of genes associated with these sites (right). The number of clusters was defined by kmean (*k* = 2) in H3K27ac enriched peaks. All analyses were performed in duplicate. **e** Known motif scanning of the sites enriched with H3K27ac modification in the gene promoter regions identified the top 10 significantly enriched motifs (blue dots) and motifs of interferon-regulatory factors (IRF) or interferon-sensitive response element (ISRE) (red dots). **f** Increased acetylation of *CD274* gene regulatory elements including an upstream enhancer (red line region) and the proximal promoter region (blue line) in FOXA1-KO versus parental UM-UC-1 cells (top). For reference, unchanged peaks are shown (grey line). FOXA1 binding sites were also identified in the promoter region of *CD274* gene by ChIP-Seq (bottom).

Fig. 11). Additionally, of the 24 TCGA cancer types, an inverse association between *FOXA1* and *CD274* gene expression was most significant in bladder cancer (Fig. 5h).

### *FOXA1* knockout induces genome-wide epigenetic reprogramming and increased expression of PD-L1 and other interferon-stimulated genes

To identify the genes dysregulated by FOXA1 loss in a bladder cancer context, we next performed ChIP-Seq analysis of the parental UM-UC-1 and FOXA1-KO isogenic cells. This analysis identified 21,766 genomic loci occupied by FOXA1, most of which were annotated as genomic enhancers (n = 13,972, 64%, defined as 50 kb upstream of Transcriptional Start Site (TSS), or 5 kb downstream of Transcriptional End Site [TES]). Intergenic regions were the second most common loci (n = 6074, 28%), followed by promoter sites (n = 1720, 8%, 2 kb up- or down-stream of TSS) (Fig. 6a). Genome-wide changes in histone H3 lysine 27 acetylation (H3K27ac), a marker of active enhancers and promoters, has been reported in several FOXA1 dysregulated cancer

types[50–56]. We, therefore, hypothesized that increased expression of PD-L1 and other interferon-stimulated genes (ISGs) in bladder cancer cells following FOXA1 loss-of-function is partly a result of changes in H3K27ac modification. To explore this possibility, we performed a comprehensive analysis of FOXA1-occupied sites and investigated changes in H3K27ac modification following FOXA1 knockout, and correlated these changes with changes in gene expression profiles. H3K27ac ChIP-Seq analysis identified 14,977 genomic regions with H3K27ac enrichment in either parental UM-UC-1 or FOXA1-KO cells (FDR < 0.05). The majority of these H3K27ac regions mapped to enhancer (n = 9302, 62%) and intergenic (n = 3064, 21%) regions and less frequently to proximal promoter regions (n = 2605, 17%) (Supplementary Fig. 12a). Together, 16% (4778) of the genomic regions were identified as having both H3K27ac enrichment and an occupied nearby (within 5 kb) FOXA1 binding site. Additionally, 55% and 29% of genomic regions were associated with either an occupied FOXA1 binding site or H3K27ac enrichment, respectively (Fig. 6b). The sites exhibiting both H3K27ac enrichment and an occupied FOXA1 binding

site (16%) were proportionally annotated as enhancer (67%), intergenic (23%) and promoter regions (10%) (Fig. 6c).

Among the H3K27ac peaks enriched within enhancer and intergenic regions, we identified 5,104 sites located within 10 kb from the gene transcriptional start site. By clustering analysis, there were 2083 (cluster 1) and 1755 (cluster 2) regions demonstrating significant increases or decreases in H3K27ac following FOXA1-KO, respectively (Fig. 6d, left). Enrichment in H3K27ac was also positively associated with increased expression of nearby genes (Fig. 6d, right). Both cluster 1 and cluster 2 exhibited evidence of FOXA1 binding (Fig. 6d, middle), suggesting that changes in FOXA1 binding at genomic enhancer regions was associated with both increased and decreased H3K27ac and parallel changes in target gene expression. Interestingly, a similar analysis of promoter regions showed that H3K27ac modification was enriched at promoter regions in FOXA1-KO cells and associated with increased gene expression (Supplementary Fig. 12b).

As H3K27ac can be a marker for chromatin regions with active regulatory elements[57,58], we next performed motif analysis to identify transcription factor binding sites associated with these enriched H3K27ac modifications. In support of an epigenetic role for FOXA1 in accessibility of ISGs, interferon-sensitive response elements (ISRE) were identified in promoter regions with H3K27ac modifications ($q < 0.05$), as well as motifs for interferon response factors (IRF1, IRF2, and IRF3; all $q < 0.05$) (Fig. 6e). Notably, three ISRE motifs were identified in the proximal promoter region of the *CD274* gene (Supplementary Fig. 12c) and these three ISRE motifs were associated with both a FOXA1-occupied region in parental UM-UC-1 and an enriched H3K27ac modification in FOXA1-KO cells (Fig. 6f). Similarly, *cis*-regulatory regions with increased H3K27ac following *FOXA1* KO also overlapped with FOXA1 bound sites in parental UM-UC-1 for several other ISGs, including *IFIT2*, *IFIT3*, *IFI35* and *STAT2* (Supplementary Fig. 12d). In sum, the results suggest that increased H3K27ac enhances accessibility of interferon response factors following genetic ablation of *FOXA1*, resulting in increased ISG expression.

### FOXA1 knockout increases IRF1 expression and its binding to the *CD274* promoter, and upregulates PD-L1 expression

Our results show that *FOXA1* KO results in increased PD-L1 expression, as well as increased H3K27ac at the *CD274* promoter. We also show that at the *CD274* promoter, and throughout the genome, areas of increased H3K27ac were significantly enriched for ISRE motifs following *FOXA1* KO. Focusing on *CD274* as a model ISG, we hypothesized that FOXA1 serves as a repressor of *CD274* in the absence of IFNɣ exposure. Since IRF1 is a key transcriptional activator of *CD274* following IFNɣ stimulation, we additionally hypothesized that IRF1 competes with and displaces FOXA1 after treatment with type II interferon. To explore these hypotheses, we performed DNA-affinity purification of nuclear proteins from UM-UC-1 cells treated in the absence or presence of IFNɣ (Fig. 7a, b). As a DNA probe, we used a 30 bp 5′ biotinylated fragment of the human *CD274* promoter, including two functional ISRE motifs, overlapping with the identified FOXA1 binding motif. Importantly, this FOXA1 bound region shares sequence homology with a FOXA1 binding region of mouse *Cd274*[48]. In addition to this wild-type probe, a "scrambled" probe and a probe with mutations in the nucleotides hypothesized to be important for FOXA1 binding within the ISRE element were used as controls (see materials and methods and Supplementary Fig. 14). DNA affinity purification experiments followed by western blotting show that in the absence of IFNɣ, FOXA1 exhibits significantly greater binding to the wild-type *CD274* probe relative to scrambled (Student's *t*-test; $p = 0.0319$) or FOXA1 mutant (Student's *t*-test; $p = 0.0256$) *CD274* probes (Fig. 7b). In the presence of IFNɣ, FOXA1 still showed significantly greater binding to wild-type *CD274* promoter relative to scrambled (Student's *t*-test; $p = 0.0208$) and FOXA1 mutant (Student's *t*-test; $p = 0.0123$) probes (Fig. 7b). However, there were no significant differences in FOXA1

binding relative to IFNɣ treatment. While FOXA1 was not regulated via IFNɣ treatment, IRF1 was significantly upregulated following IFNɣ stimulation (Fig. 7a), resulting in increased IRF1 binding to wild-type *CD274* relative to scrambled and mutant probes (Fig. 7b). These results indicate IRF1 does not compete with FOXA1 binding to the *CD274* promoter following IFNɣ stimulation.

Interestingly, we noted that *FOXA1* KO resulted in significant increases in IRF1 transcript (Fig. 7c; Student's *t*-test; $p < 0.0001$) and protein levels (Fig. 7d). Based on this, we tested the hypothesis that increased IRF1 expression following *FOXA1* KO results in increased binding of IRF1 to the *CD274* promoter, and subsequently increased *CD274* expression. Indeed, DNA affinity purification from *FOXA1* KO UM-UC-1 cells identified an ~26-fold increase in IRF1 binding to the wild-type *CD274* promoter relative to parental UM-UC-1 (Fig. 7e, f). To determine if increased IRF1 expression was sufficient to drive expression of *CD274* and other ISGs even in the absence of IFNɣ stimulation, we next transiently overexpressed IRF1 in parental UM-UC-1 cells (Fig. 7g, h). Ectopic overexpression of IRF1 resulted in a significant increase in the expression of *CD274* (Fig. 7i; Student's *t*-test; $p = 0.0344$), *STAT2* (Fig. 7j; Student's *t*-test p = 0.0321), *ISG15* (Fig. 7k; Student's *t*-test $p = 0.0008$), *IFIT2* (Fig. 7l; Student's *t*-test; $p = 0.0338$), *IFIT3* (Fig. 7m; Student's *t*-test; $p = 0.0009$), and *IFI35* (Fig. 7n; Student's *t*-test; $p = 0.0406$). In sum, these results show that increased IRF1 expression is sufficient to activate the expression of ISGs, including *CD274* in an IFNɣ-independent manner.

## Discussion

Elucidating the relationship between genomic and immune heterogeneity is crucial to understanding mechanisms of intrinsic and acquired resistance to immune checkpoint blockade in patients with bladder cancer and other cancer types. Here, we performed comprehensive multiplatform analyses of spatially distinct regions of urothelial and squamous morphology from mixed histology bladder cancers. While WES confirmed that both the UC and SqD regions of individual tumors were derived from a shared precursor, we observed significant mutational discordance between the two morphologically distinct regions consistent with early branched evolution. Of clinical significance, heterogeneous histomorphology was a marker for immune heterogeneity within the tumor microenvironment and differential PD-L1 expression on tumor cells, and patients whose bladder tumors had marked intratumoral morphologic heterogeneity were less likely to benefit from treatment with the anti-PD-L1 inhibitor atezolizumab. Functional studies indicated that lineage plasticity and immune cell heterogeneity were coordinately dysregulated through changes in the expression of the FOXA1 transcription factor, the loss of which was sufficient to induce PD-L1 expression on tumor cells via increased IRF1 expression and its binding to the CD274 promoter.

While we did not identify a recurrent genetic alteration unique to the squamous regions of mixed histology tumors, the squamous components generally exhibited greater divergence from their nearest shared evolutionary ancestor and greater tumor mutational burden, neoantigen load and ploidy than the matched UC regions. The molecular basis for the higher mutational burden of the SqD regions was not clear as in most cases, the paired UC and SqD regions demonstrated similar mutational processes with almost all cases exhibiting APOBEC-associated mutational signatures[13,18]. Our analysis also identified discordant mutations in genes long thought to represent early clonal events during bladder carcinogenesis such as *TP53*, *RB1*, and *FGFR3*, the latter being unique to the SqD regions of two cases.

Despite the presence of distinct morphologic differences between SqD and UC regions of individual tumors, all twelve SqD regions and nine of twelve UC regions analyzed by RNAseq were basal-squamous subtype based on the TCGA molecular classification schema, with the remaining three UC tumors being luminal or luminal-papillary subtype. Our bulk multi-region and single cell RNA sequencing suggested that in

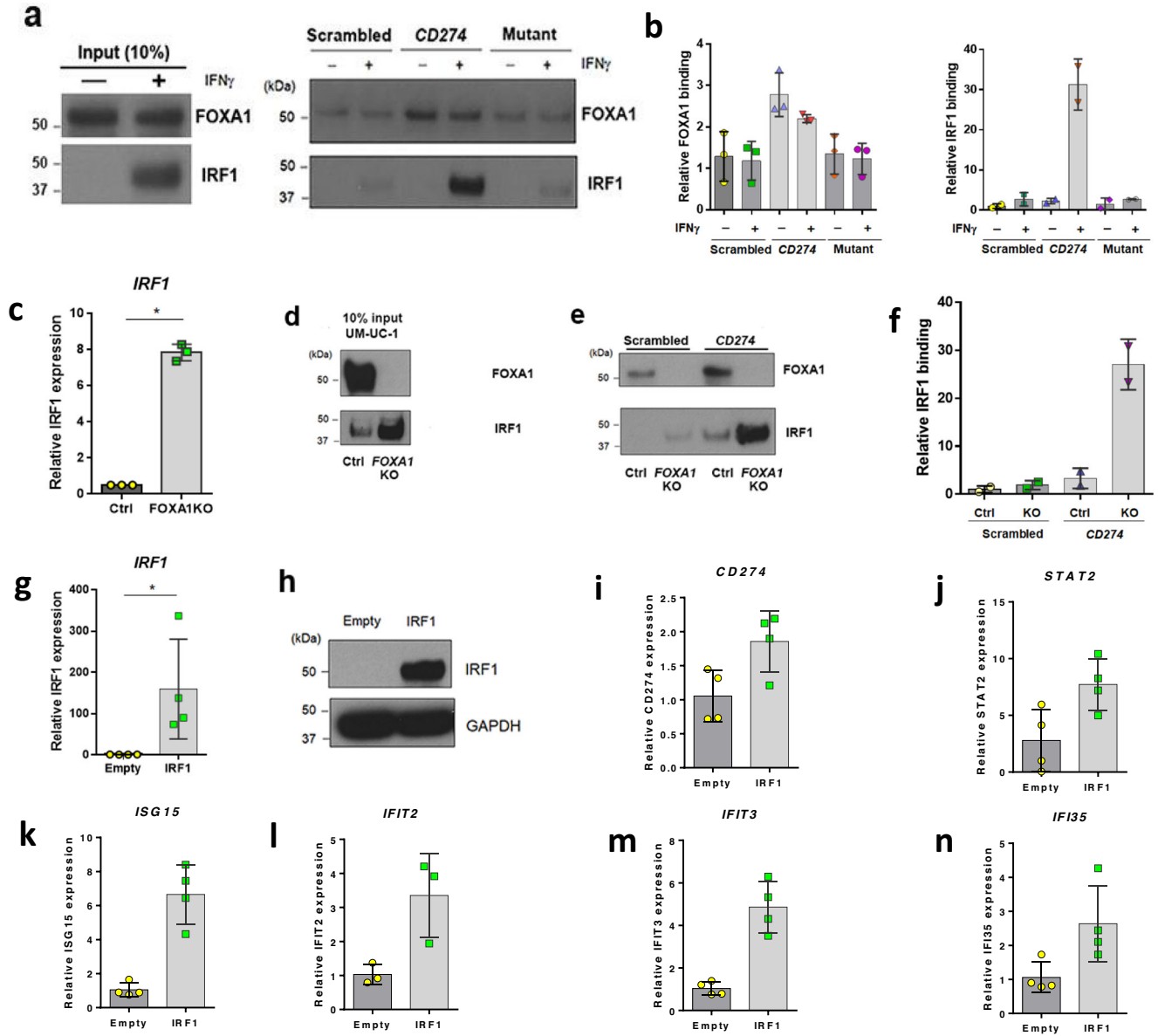

**Fig. 7 | FOXA1 KO increases expression of IRF1 and enhances IRF1 binding to the _CD274_ promoter.** Western blotting (**a**) and quantification (**b**) following DNA affinity purification for FOXA1 (n = 3) and IRF1 (n = 2) in parental UM-UC-1 treated in the presence and absence of IFNɣ. In the absence of IFNɣ, FOXA1 exhibits significantly greater binding to the wild-type _CD274_ probe relative to scrambled negative control DNA probe (Student's _t_-test; p = 0.0319) or mutant (Student's _t_-test; p = 0.0256) _CD274_ probes. In the presence of IFNɣ, FOXA1 still showed significantly greater binding to wild-type _CD274_ promoter relative to scrambled DNA probe (Student's _t_-test; p = 0.0319) and FOXA1-binding mutant (Student's _t_-test; p = 0.0256) _CD274_ probes. There were no significant differences in FOXA1 binding relative to IFNɣ treatment. Following IFNɣ treatment, IRF1 shows increases in binding to wild-type _CD274_ relative to scrambled and mutant probes. Q-RT-PCR data are expressed as the mean ± S.D. from independent experiments of FOXA1 (n = 3), IRF1 (n = 2), respectively. Source data are provided as a Source Data file (**c**) and western blotting (**d**) shows that _FOXA1_ KO in UM-UC-1 results in significant increases in IRF1 expression at the mRNA and protein levels, respectively. Data of Q-RT-PCR are

expressed as the mean ± S.D. from independent experiments (n = 3). Source data are provided as a Source Data file. Western blotting (**e**) and quantification (**f**) following DNA affinity purification for FOXA1 and IRF1 in parental UM-UC-1 (Ctrl) and _FOXA1_ KO UM-UC-1 (_FOXA1_ KO) cell lines. IRF1 purified from _FOXA1_ KO UM-UC-1 exhibited a 26-fold increase in binding to the _CD274_ promoter fragment relative to parental UM-U-C1. IRF1 was unable to be purified from parental and _FOXA1_ KO UM-UC-1 cells with scrambled negative control oligo (n = 2). Ectopic expression of IRF1 in parental UM-UC-1 cells followed by Q-RT-PCR (n = 4). Data are expressed as the mean ± S.D. from independent experiments (**g**) and western blotting (**h**). Source data are provided as a Source Data file. Overexpression of IRF1 significantly increased expression of **i** _CD274_ (n = 4, Student's _t_-test; p = 0.0344), **j** _STAT2_ (n = 4, Student's _t_-test p = 0.0321), **k** _ISG15_ (n = 4, Student's _t_-test p = 0.0008), **l** _IFIT2_ (n = 3 Student's _t_-test; p = 0.0338), **m** _IFIT3_ (n = 4, Student's _t_-test; p = 0.0009) and **n** _IFI35_ (n = 4, Student's _t_-test; p = 0.0406). Data are expressed as the mean ± S.D. from independent experiments. *p < 0.05 was considered as a statistically significant. ns not significant. All tests are unpaired two-sided Student's _t_-test.

urothelial carcinomas with SqD, lineage plasticity with acquisition of squamous features is a continuous and gradual process in which the squamous morphology represents the extreme of the basal-squamous phenotype. More specifically, subpopulations of cancer cells from the SqD regions recapitulated the spectrum of differentiation stages present in normal epidermis. This was best shown using single cell analysis

which was able to identify distinct cell populations with expression profiles characteristic of basal, suprabasal, spinous, granular and cornified squamous cells. Given that the expression profiles of the UC regions of most mixed histology tumors were basal-squamous, our data indicate that the process of SqD is well underway at the molecular level even in regions lacking characteristic squamous

histomorphologic features. Overall, these results highlight a limitation of current tumor classification schema which are largely reliant on histologic features identifiable under light microscopy. It is widely acknowledged that the UC regions of bladder tumors with SqD are typically morphologically indistinguishable under light microscopy from non-keratinizing squamous cell carcinomas arising in organs that have a squamous lining, such as squamous tumors of the head and neck and lung. However, in contrast to tumors arising at these other anatomic sites, the squamous designation is only applied to bladder cancers when there is definitive evidence of keratinization or intercellular bridges (representing desmosomes)[59]. This deliberately restrictive tumor classification approach was adopted to minimize overcalling of squamous differentiation in tumors that resemble the UC component of mixed histology bladder cancers with UC and SqD features. Our data suggest thus that molecular analysis, in particular gene expression profiling, is better suited than histologic examination under light microscopy for classifying bladder cancers with potential implications for treatment selection and patient outcomes.

A notable finding of our gene expression analyses was that UC regions of mixed histology tumors with SqD consistently expressed higher levels of *FOXA1*, *GATA3*, and *PPARG*, transcription factors previously shown to regulate luminal gene expression[20]. In addition to its putative role in lineage plasticity, we identified FOXA1 as a tumor cell-intrinsic repressor of ISG expression, exemplified by PD-L1. Using isogenic bladder cancer cells in which FOXA1 was genetically ablated using CRISPR/Cas9, we were able to show that loss of FOXA1 expression was sufficient to induce PD-L1 expression in bladder cancer cells.

Our results support a role for FOXA1 as a pioneer factor that regulates chromatin structure[60–62], and our findings suggest that global epigenetic reprogramming following FOXA1 loss-of-function contributes to the IFNγ-dominant signature found in the squamous regions of mixed histology bladder cancers. In our integrated analysis of RNAseq and ChIP-seq data, we also observed that genome-wide epigenetic reprogramming following loss of FOXA1 expression was associated with increased H3K27ac at regulatory elements important for the control of *CD274* and other interferon sensitive genes. These areas of increased H3K27ac after *FOXA1* knockout were significantly enriched for ISRE motifs. Our studies also provide evidence that increased IRF1 expression following *FOXA1* KO increases expression of *CD274* and other ISGs in an interferon-independent manner, providing one mechanistic explanation of how *FOXA1* loss increases expression of *CD274* and other ISGs. Previous work published during revision of this manuscript reported that FOXA1 physically interacts with STAT1 and STAT1/STAT2 following treatment with IFNγ and IFNα, respectively[63]. Furthermore, this work showed that FOXA1 interaction with STAT proteins serves to repress IFNα-induced gene expression. Our work has identified an additional and likely complementary interferon-independent role for FOXA1 and IRF1 in the activation of *CD274* and other ISGs in epithelium. The data also provide a direct link between the molecular mechanisms mediating lineage plasticity in bladder cancer and changes in the composition of immune cells within the tumor microenvironment. While FOXA1 has previously been implicated in the positive regulation of PD-L1 expression in a subset of immune cells[48], here we show that FOXA1 also functions as a tumor cell-intrinsic regulator and repressor of PD-L1 expression in a solid tumor (namely urothelial carcinoma). Moreover, our analysis of TCGA data suggests that FOXA1 may also function as a regulator of PD-L1 expression in additional cancer subtypes including squamous cancers of the lung and head and neck.

One limitation of the current study was the sample size. This was in part due to our decision to restrict the analysis to tumors in which urothelial and squamous regions could be separated by macrodissection and analyzed individually using bulk sequencing methods. Much more commonly, urothelial and squamous regions are intermixed in a manner that precludes a paired analysis using bulk RNA and

DNA sequencing methods. The recent development of robust methods for global gene expression profiling at the single cell level has opened the possibility of analyzing a broader range of mixed histology tumors. As these methods typically require fresh and not frozen tissue, we have initiated efforts to prospectively perform integrated single cell analysis of mixed histology bladder tumors. While we focused only on tumors with SqD in this study, there are other histologic variants of UC that warrant further investigation, in particular those that are associated with an aggressive clinical course, such as micropapillary, plasmacytoid, small cell and sarcomatoid carcinomas[11,12,64,65]. While patients with such tumors are often excluded from clinical trials of novel agents such as immune checkpoint inhibitors, the recent FDA-approvals of anti-PD-L1 and anti-PD1 antibodies for patients with bladder cancer should facilitate future biomarker studies to determine how the lineage plasticity observed in other histologic variants impacts the tumor microenvironment.

In conclusion, we performed a multiplatform, integrated analysis of macrodissected UC and SqD regions from mixed histology bladder cancers with SqD. We found that lineage plasticity in bladder cancers with SqD was associated with loss of expression of the *FOXA1*, *GATA3*, and *PPARG* transcription factors. Furthermore, we identified a mechanistic link between lineage plasticity in these tumors and changes in the immune tumor microenvironment. PD-L1 expression on tumor cells was highly discordant in the urothelial and squamous regions. As anti-PD1/PL-1 antibodies are being adopted as standard therapy for patients with locally advanced and metastatic bladder cancer, and as PD-L1 expression is associated with response to these therapies, the work presented here has immediate translational implications for the identification of bladder cancer patients most likely to respond to immune checkpoint blockade. Our data also suggest that morphologic heterogeneity, as a biomarker of genomic and immune heterogeneity, should be assessed for in ongoing and future clinical trials of novel immunotherapy combinations and validated prospectively as a biomarker of intrinsic resistance to immunotherapy across cancer types.

## Methods

### Patient samples and immunohistochemistry (additional details provided in Supplementary Table 1)

All specimens were obtained from patients following written informed consent and in accordance with institutional review board (IRB) approval at Memorial Sloan Kettering Cancer Center (IRB# 06-107 and IRB# 89-076), Pennsylvania State University College of Medicine (IRB# STUDY00000620) and Vanderbilt University Medical Center (IRB#140888). Tumor samples were from surgical specimens (cystectomy or transurethral resection specimens) and were classified as urothelial carcinoma, NOS (UC), or Squamous Differentiation (SqD) by consensus of three board certified genitourinary pathologists (H.A., J.I.W., and L.L.G.). Tumors with regions of variant histology other than SqD were excluded from this study. Macrodissection was performed with the assistance of hematoxylin and eosin staining to isolate areas of UC or SqD. Blood and/or normal tissues (benign lymph nodes procured at the time of cystectomy) were used as a source of germline DNA. Immunohistochemistry for GATA3 (L50-823, Biocare), FOXA1 (HNF-3α/β [C-20]: sc-6553, Santa Cruz), PPARG (D69, Cell Signaling), and PD-L1 (SP263, Ventana) was performed using the antibodies specified. In total, 42 tumor regions of the primary bladder tumor from 21 patients (one UC and one SqD region per tumor), from 16 male and 6 female patients, age range 52–83 years, underwent whole-exome sequencing (WES). Among the 21 tumors, 12 UC-SqD pairs were of sufficient quality for whole transcriptome profiling (RNA-Seq). One additional tumor sample from a male patient was subjected to single-cell RNA-Seq. The gender distribution in this cohort is consistent with the overall disease prevalence (bladder cancer is 3–4 times more common in men than in women)[13].

## Whole exome sequencing

Whole exome sequencing was performed by the MSKCC Center for Molecular Oncology (CMO). Briefly, DNA was extracted using the DNeasy Blood & Tissue Kit (Qiagen, Valencia, CA) according to the manufacturer's modified protocol and captured using Human whole exome Sureselect (Agilent). Libraries were then sequenced on a HiSeq2500 using 100 bp pair-end mode. The average numbers of read pairs per sample was 78, 99, and 105 million for the normal, UC, and SqD regions, respectively. Average duplication rates were 0.08%, 0.23%, and 0.35% for normal, UC, and SqD, respectively. A comprehensive data analysis pipeline (TEMPO, https://ccstempo.netlify.app), developed by the MSK Center for Molecular Oncology (CMO) team, was used to perform read alignment, variant calling, mutational signature analysis, copy number and clonality analysis, neoantigen prediction, microsatellite instability analysis.

For **clonality** analysis, the Clonality R package (v1.40.0)[16] was applied to test whether a pair of tumors from the same patient were clonal (derived from a common precursor) or independent (distinct primary tumors). When analyzing somatic mutation data, as was performed here, the Clonality R package applies a statistical test developed by Ostrovnaya et al.[17]. In brief, a conditional likelihood model is applied to test the null hypothesis that the tumors are independent. The model only uses loci where at least one of the two tumors has a mutation. Marginal mutation frequencies are estimated from an external reference, and in this analysis the frequencies were derived from mutations observed in the TCGA BLCA cohort[13]. A generalized likelihood ratio test is performed based on the number of observed mutations common to both tumors or those that appear in exactly one of the two tumors. The Clonality output is presented in Supplementary Table 2.

For **phylogeny** analysis, evolutionary relationships of mutation phylogeny between UC and SqD regions were inferred using the union of somatic mutations called in any of the paired samples, a normal sample with none of these mutations was added for each patient. Package "Analyses of Phylogenetics and Evolution" (R package "ape", v5.4-1) was used to infer the phylogenetics relationship based on the cluster results of the mutation table of the UC, SqD and the assumed normal sample.

## RNA-Sequencing

RNA from formalin fixed paraffin embedded tumor tissue was extracted using the RNeasy mini kit (Qiagen; Valencia, CA) according to manufacturer's instructions. After ribogreen quantification and quality control using a Agilent BioAnalyzer, 500 ng to 2 μg of total RNA underwent polyA selection and Truseq library (TruSeq™ RNA Sample Prep Kit v2) preparation. Briefly, samples were fragmented for 2 min at 94 °C before undergoing first strand and second strand cDNA synthesis. Libraries were amplified with 10 cycles of PCR and size-selected for fragments between 400 and 550 bp with a Pippin prep instrument (Sage Science). Samples were barcoded and run on a Hiseq 2500 in a 50 bp paired end run, using the TruSeq SBS Kit v3 (Illumina). Sequence reads were aligned and counted for each gene using the RSEM algorithm (v1.2.25) with the STAR alignment program (v2.5.0)[66]. An average of 30 million paired reads were generated per sample. Reads normalization and differentially expressed gene were analyzed using DESeq2 (v1.30.0)[67]. Gene pathway analysis was performed using GSEA (v.2.2.0)[68]. Immune cell fractions were inferred using both the GSVA (v1.44.2)[68] and CIBERSORTX (v1.0)[69] programs.

## Single-cell RNA sequencing

For single-cell RNA sequencing, tumor tissue was processed immediately after removal from the bladder. Briefly, tumor tissue was cut into small pieces (<1 mm in greatest dimension) and incubated in 1 ml media supplied with enzymes from the tumor dissociation kit (Miltenyi, Cat: 130-095-929) for 30 min on a 37 °C shaker. Subsequently,

10 ml DMEM media was added to dilute the suspension, then a 40-μm cell mesh was used to filter the suspension. After centrifugation at 250 g for 5 min, the supernatant was discarded, and the cells were washed with DMEM media twice. The pellet was resuspended in 1 ml of calcium- and magnesium-free PBS containing 0.04% weight/volume BSA. Finally, 10 μl of suspension was counted under an inverted microscope with a hemocytometer. Trypan blue was used to quantify live cells.

Following the manufacturer's protocol, the Chromium Single cell 3′ Reagent v3 kit was used to prepare barcoded scRNA-seq libraries. Single-cell suspensions were loaded onto a Chromium Single-Cell Controller Instrument (10x Genomics) to generate single-cell gel beads in emulsions (GEMs). To capture 10,000 cells per library, approximately 15,000 cells were added to each channel. After generation of GEMs, reverse transcription reactions were engaged to generate barcoded full-length cDNA. Next, cDNA was amplified, fragmented, end-repaired, A-tailed, and ligated to an index adaptor, and then the library was amplified. The library was sequenced on a HiSeq X Ten platform (Illumina) to generate 150 bp paired-end reads.

Sequencing reads were aligned to GRCh38 by Cell Ranger (10x genomics, v3.0) and analyzed with the Seurat package v4.0[70]. Cells with UMI numbers <1000 or with over 10% mitochondrial-derived UMI counts were considered low-quality cells and were removed. Doublets were predicted using DoubletFinder, v2.0 and excluded after which 4768 cells remained for downstream analysis. Data normalization, dimension reduction (RunUMAP), and cluster identification (FindClusters) were performed with default parameters. To map differentiation of the tumor cell subpopulations, pseudotime analysis was performed using the Slingshot package in R[71] with default parameters.

## Expression and immune subtype assignment

Macrodissected areas of SqD and UC were assigned to TCGA molecular subtypes by nearest centroid analysis using the BLCAsubtyping package in R [https://github.com/cit-bioinfo/BLCAsubtyping] (v. 2.1). Immune subtypes were assigned by nearest centroid analysis and the ClaNC package (v1.1). For the immune subtype portion of the analysis, we randomly selected 1000 non-bladder cancer cases from the TCGA dataset. Subtypes for these had been assigned in the study by Thorsson et al., supplemental data[42]. We then generated centroids for the C1-C4 immune subtypes from these cases using RNA sequencing data applied to ClaNC, with each centroid defined by 500 genes. SqD and UC areas separated by macrodissection were assigned to the nearest centroid, using correlation-based distance.

## ChIP-Seq

Cells were fixed with 1% formaldehyde for 15 min and quenched with 0.125 M glycine. Chromatin was isolated by the addition of lysis buffer, followed by disruption with a Dounce homogenizer. Lysates were sonicated, and the DNA sheared to an average length of 300–500 bp. Genomic DNA (Input) was prepared by treating aliquots of chromatin with RNase, proteinase K, and heat for de-crosslinking, followed by ethanol precipitation. Pellets were resuspended, and the resulting DNA was quantified on a NanoDrop spectrophotometer. Extrapolation to the original chromatin volume allowed quantitation of the total chromatin yield. An aliquot of chromatin (30 μg) was precleared with protein A agarose beads (Invitrogen). Genomic DNA regions of interest were isolated using 4 μg of antibody against H3K27ac (Active Motif, cat# 39133) or FOXA1 (Abcam, cat# ab5089). Complexes were washed, eluted from the beads with SDS buffer, and subjected to RNase and proteinase K treatment. Crosslinks were reversed by incubation overnight at 65 °C, and ChIP DNA was purified by phenol-chloroform extraction and ethanol precipitation. Quantitative PCR (QPCR) reactions were carried out in triplicate on specific genomic regions using SYBR Green Supermix (Bio-Rad). The resulting signals were normalized for primer efficiency by carrying out QPCR for each primer pair

using Input DNA. Illumina sequencing libraries were prepared from the ChIP and Input DNAs by the standard consecutive enzymatic steps of end-polishing, dA-addition, and adaptor ligation. Steps were performed on an automated system (Apollo 342, Wafergen Biosystems/Takara). After a final PCR amplification step, the resulting DNA libraries were quantified and sequenced on an Illumina NextSeq 500 (75 nt reads, single end). Reads were aligned to the human genome (hg38) using the Bowtie2 algorithm (default settings). Quality and adapter trimming of raw data was performed using trim galore (v0.6.5). Duplicate reads were removed by PICARD MarkDuplicate and only uniquely mapped reads (mapping quality >= 25) were used for further analysis (average deduplicated 23 million reads per sample). Peak locations were determined using the MACS2 algorithm (v2.1.0) with a cutoff of $p$-value = 1e−7. Peaks that were on the ENCODE blacklist of known false ChIP-Seq peaks were removed. Differentially modified H3K27ac loci were identified by the DiffBind package (v3.0.15). Peak annotation was performed using the Homer program (v4.11). Promoter region cis-element prediction was performed using the universal motif package (v1.6.3).

### Immunofluorescence staining
Multiplex immunofluorescence (mIF) was performed for the following markers: PD-L1 (1:400, E1L3N, Cell Signaling), CD4 (1:200, EPR6855, abcam), CD8 (1:400, C8/114B, Cell Signaling), CD56 (1:2, MRQ-42, Cell MARQUE), CYP27A1 (1:600, EPR7529, abcam), and p63 (1:200, D9L7L, Cell Signaling). The tissue was de-paraffinized and underwent heat-mediated antigen retrieval in citrate buffer prior to triple labeling with the Opal Multiplex Immunostaining kit (Perkin Elmer, Waltham, MA). Briefly, slides were rinsed with TBST for 10 min, incubated in 1% bovine serum albumin for 30 min at room temperature (RT), and incubated with the anti-CD4 primary antibody for 2 h followed by washing with TBST (3 × 2 min). Slides were incubated with secondary antibody (Genomic technology co. LTD, Shanghai, China) for 30 min at RT, washed with TBST (3 × 2 min), incubated with the Opal 520 fluorophore working solution (1:300) for 10 min at RT, and washed with TBST (3 × 2 min). Slides then underwent heat-mediated antigen retrieval and antibody removal in citrate buffer. After cooling, the process was repeated for two additional rounds for labeling with anti-CD8, CD56, CYP27A1, PD-L1, and p63, followed by secondary labeling with the respective fluorophore working solution [i.e, Opal 570 (1:300), Opal 650 (1:500), Opal 690 (1:300) and Opal 620 (1:1000), respectively, (PerkinElmer, Waltham, MA)] as described above. Next, nuclei were labeled with DAPI for 10 min in a humid chamber, washed with dH$_2$O, and coverslips were mounted with glycerin. Upon completion of multiplex IF staining, the slides were imaged using the Vectra 3.0 spectral imaging system (Perkin-Elmer). The chromogenic IHC-stained slides were scanned by using the bright field protocol, and the uniplex and multiplex IF staining was imaged by using the fluorescence protocol at 10 nm λ from 420 nm to 720 nm, to extract fluorescent intensity information from the images. A similar approach was used to build the spectral library using the InForm 2.2.1 image analysis software (PerkinElmer).

### Cell culture and Western Blot analysis
Human bladder cancer cells (UM-UC-1, UM-UC-3) were purchased from ATCC, and authenticity was confirmed by short tandem repeat (STR) analysis[20] or by MSK-IMPACT analysis[72]. All cell lines are routinely screened for mycoplasma. Cells were cultured in Minimal Essential Medium (UM-UC-1, UM-UC-3) supplemented with 10% FBS. Western blotting was performed as described previously[20]. Primary antibodies were used as follows: FOXA1 (1:500, ab23738, Abcam), PD-L1 (1:1000, ab213524, abcam), GAPDH (14C10) (1:1000, #2118, Cell signaling).

To establish UM-UC-1 FOXA1 knockout (KO), UM-UC-1 ($2 \times 10^5$) cells were transfected with 2.5 mg of HNF-3alpha CRISPR/Cas9 KO plasmid (Santa Cruz, sc-400743) using lipofectamine3000 (Thermo fisher scientific). Three GFP-positive cells were sorted in a single well of 96-well plate containing 100 ml of medium by Aria II cell sorter (BD Biosciences). Sorted cells were expanded and knockout of *FOXA1* in UM-UC-1 cells was confirmed by western blot analysis.

### Transient transfection
One day before transfection, $2 \times 10^5$ UM-UC-3 or UM-UC1 cells were plated in wells of 6 well plates. The next day, attached cells were transfected with pCMV6-Entry (Origene; CMV empty vector) or pCMV6-FOXA1 (Origene; RC206045) or pCMV6-IRF1 (Origene; RC203500) using lipofectamine3000. After 48 h, RNA and protein were collected using the RNeasy kit (Qiagen) and lysed using RIPA buffer (Thermo fisher scientific) containing protease inhibitor (Roche) according to manufacturer's protocol.

### Regulon analysis
We inferred the FOXA1 regulons using the *RTN* package (v2.13.2)[73,74]. We estimated single-sample regulon activity by a two-tailed gene set enrichment analysis (GSEA-2T), which was used to sort samples of the TCGA BLCA[13] and Sjödahl et al.[49]. cohorts. The GSEA-2T was performed in $R$[75] using the *RTN*[73] and *RTNsurvival* packages (v1.20.0)[76]. For the Kaplan-Meier analysis, we stratified the cohort into 2 groups—positive and negative regulon activity—and evaluated differences between the groups using a Logrank test.

### Gene expression correlation analysis
Gene expression data of each cancer cohort in the Cancer Genome Atlas (TCGA) project were analyzed by TCGAbiolinks package (v2.24.3). The expression correlations between CD274 and FOXA1 were calculated based on the normalized reads using cor.test function in R with Spearman method. The results of correlation factors and $p$-values were summarized, and Bonferroni adjusted p-values were calculated by p.adjust function.

### DNA pulldown assay
DNA pulldown assays were performed similar to as described in ref. 77, with the following modifications. 5′-biotinylated probes used corresponded to a region of the promoter of the human *CD274* gene, which includes FOXA1 and ISRE motifs. The following 5′ biotinylated DNA probes were used in this study. Wild-type CD274 (forward 5′-ACTGA-CATGTTTCACTTTCTGTTTCATTTC; reverse 5′-GAAATGAAACAGAA AGTGAAACATGTCAGT), Mutant (forward 5′-ACTGACACGTCGCAC CGACCGTCGCACGAC; reverse 5′-GTCGTGCGACGGTCGGTGCGACG TGTCAGT), scrambled (forward 5′-CGAGCGATCGAGCGATCGAGC-GATCGAGCG; reverse 5′-CGCTCGATCGCTCGATCGCTCGATCGCT CG). DNA probes were annealed in annealing buffer (10 mM Tris (pH8.0), 50 mM NaCl, 1 mM EDTA (pH8.0)) at 95 C for 5 min, and then cooled to room temperature. Annealed probes were then added to streptavidin-coated beads (Thermo Fisher Scientific). Probe-conjugated beads were incubated in blocking buffer (20 mM Tris (pH8.0), 15% glycerol, 0.05% NP40) containing 50 mg of sonicated salmon sperm DNA. Beads were washed 3 times with 200 ml of blocking buffer (no salmon sperm DNA) and incubated at 4 C overnight with nuclear lysates. Nuclear lysates of UM-UC1 cells, UM-UC1 cells after 18 h treatment with IFNγ (100 ng/ml) (R&D systems) or vehicle control (distilled water), or *FOXA1* KO UM-UC1 cells were prepared using NE-PER Nuclear and Cytoplasmic Extraction Reagents (Thermo Fisher Scientific). After incubation, beads were washed 3 times with 200 µl of blocking buffer without salmon sperm DNA. After washing, bound proteins were eluted with 50 µl of elution buffer (1xNuPAGE LDS sample buffer, 10% mercaptoethanol). Samples were

run on 4–12% NuPAGE gel (Thermofisher Fisher Scientific) and were immunoblotted using primary antibodies for FOXA1 (1:500, ab23738, Abcam) and IRF1 (D5E4) (1:1000, #8478, Cell signaling).

## Statistics and reproducibility

Cell line experiments were performed in triplicate at least two times. No statistical method was used to predetermine sample size. No data were excluded from the analyses. The experiments were not randomized. The Investigators were not blinded to allocation during experiments and outcome assessment.

## Reporting summary

Further information on research design is available in the Nature Research Reporting Summary linked to this article.

## Data availability

All somatic mutational calls and CNAs along with accompanying clinical data will be available for analysis and visualization in the cBioPortal for Cancer Genomics (https://cbioportal.mskcc.org/study/summary?id=blca_cmo_06155_2016). Raw whole exome sequencing data have been deposited in the Database of Genotypes and Phenotypes under dbGaP Accession phs001783.v4.p1. Due to informed consent requirements related to the genomics results uploaded to dbGaP, this data is made available through controlled-access. Data access is provided by dbGaP Authorized Access [https://dbgap.ncbi.nlm.nih.gov/aa/wga.cgi?page=login] upon request. Use of the data must be related to Cancer. Requestor agrees to make results of studies using the data available to the larger scientific community. Use of the data includes methods development research (e.g., development of software or algorithms). According to the dbGaP agreement outlines, access to the requested dataset(s) is granted for a period of one (1) year, with the option to renew access or close-out a project at the end of that year. Details on how to obtain authorized access from dbGaP can be retrieved through this link [https://www.ncbi.nlm.nih.gov/projects/gap/cgi-bin/GetPdf.cgi?document_name=GeneralAAInstructions.pdf].

The RNASeq data generated in this study (including bulk RNA-Seq and scRNA-Seq raw data) and ChIP-Seq data generated from the UM-UC-1 isogenic cells have been deposited in the GEO database under accession code GSE172433.

Source data are provided with this paper. Figures associated with raw data include Figs. 1–7 and Supplementary Figs. 2–8, 11 and 12. The source data for Figs. 5a, b, 7a, d, e, h, and Supplementary Figs. 10a–d are provided as a Source Data file.

The remaining data are available within the Article, Supplementary Information or Source Data file.

The publicly available TCGA data used in this study are available in the Genomic Data Commons database accessible thorough this link [https://gdc.cancer.gov] and in the TCGA publication page accessible through this link [https://www.cancer.gov/about-nci/organization/ccg/research/structural-genomics/tcga][13]. The Lund University (Sjödahl et al.) data used in this study are deposited at the Gene Expression Omnibus under access GSE32894[49]. The remaining data are available within the Article, Supplementary Information or Source Data file. Source data are provided with this paper.

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

## Acknowledgements

This work was supported by the Albert Institute for Bladder Cancer Care and Research (H.A.A, D.J.D.), Cycle for Survival (H.A.A, D.B.S.), the Mark Foundation (H.A.A., D.B.S.), Parker Institute for Cancer Immunotherapy (H.A.A.), the Samuel Waxman Cancer Research Foundation (Y.C., J.E.R), W81XWH-21-1-0916 from the Department of Defense (D.J.D.), RSG 17-233-01-TBE from the American Cancer Society (D.J.D.), W.W. Smith Charitable Trust (D.J.D.), Ruth Heisey Cagnoli Endowment in Urology at Penn State College of Medicine, Bladder Cancer Support Group at Penn State Health, R01 CA233899 (H.A.A.), P01CA221757 (H.A.A., D.B.S.), SPORE in Bladder Cancer P50CA221745 (H.A.A., D.B.S., D.F.B. J.E.R), the Sloan Kettering Institute for Cancer Research Cancer Center Support Grant P30CA008748, and the Marie-Josée and Henry R Kravis Center for Molecular Oncology (H.A.A., D.B.S.). The authors wish to acknowledge helpful discussions with Dr. Ryan Hobbs (Department of Dermatology, Pennsylvania State University College of Medicine) and Dr. Chris Norbury (Department of Microbiology and Immunology, Pennsylvania State University College of Medicine). The authors also wish to acknowledge Dr. Gregory Yochum (Associate Professor of Biochemistry, Pennsylvania State University College of Medicine) for providing protocols for the DNA affinity purification assay and the expert technical assistance of Dr. Yuka Imamura Kawasawa (Director, Genome Sciences Facility, Pennsylvania State University College of Medicine) and the MSK Integrated Genomics Organization (iGO).

## Author contributions

H.A.A, D.J.D, J.I.W, L.L.G, J.D.R, P.E.C., and R.A conceived the project. J.I.W, W.H, D.J.D, and H.A.A designed the study. D.J.D and H.A.A. supervised the project. J.I.W, W.H, H.Y, L.L.G, A.T.L, T.J.H, Y.L, D.J.D, and H.A.A collected data. J.I.W, W.H, H.Y, V.W, L.S, J.M.C, M.A.A.C, A.G.R (AG Robertson), F.K, I.O, T.A.C, D.J.D, and H.A.A analyzed data. W.H., V.W, F.K, I.O., and M.A.A.C. performed statistical and bioinformatic analyses. J.D.R., M.K., S.B.M., M.J., P.E.C., J.S, Y.B.C, A.G, S.J.S, S.W.F, S.K.T, K.K, J.T, N.K, S.P.G, T.N.C, T.A.C, Z.C, M.R, N.D.S, S.C, A.V, N.M, B.H.B, E.J.P, M.Y.T, G.I, J.E.R, D.F.B, M.K, S.B.M, M.J, J.A.T, V.E.R, Y.C, S.A.F, D.J.D, and H.A.A. provided clinical data and funding. J.I.W, W.H, H.Y, D.B.S, D.J.D, and H.A.A. wrote the manuscript.

## Competing interests

E.J.P. received honorarium from UpToDate, and received research funding from and is on the scientific advisory boards for Janssen Pharmaceuticals, Merck & Co. Inc., QED Therapeutics, UroGen Pharma and Steba Biotech. D.B.S. has consulted for/received honoraria from Pfizer, Loxo/Lilly Oncology, FORE Therapeutics, Vividion Therapeutics, Scorpion Therapeutics, Fog Pharma, and BridgeBio. D.J.D. received a research grant from Bristol-Myers-Squibb. H.A.A has consulted for Bristol-Myers-Squibb, AstraZeneca, Janssen Biotech, and Paige.ai. M.J. received research grant from AstraZeneca, Pfizer, and Eisai (drug only to Institution), and is on the advisory board for Seagen. J.S. is a consultant for Janssen Research & Development, LLC. S.A.F. received research support from AstraZeneca, Genentech/Roche, is a consultant/advisory board member for Merck and BioNTech, and owns stock/equity interest in Urogen, Allogene Therapeutics, Neogene Therapeutics, Kronos Bio, ByHeart, 76Bio, Vida Ventures, Inconovir, and Doximity. B.H.B is a consultant to Olympus corporation. D.F.B. reports personal fees from Bristol Myers Squibb and Merck; consulting/advisory role for Merck, Dragonfly Therapeutics, Fidia Farmaceutici S.p.A., and Bristol Myers Squibb Foundation; Travel/accommodations/expenses from Merck; and institutional research funding from Novartis, Merck, Bristol-Myers Squibb, AstraZeneca, Astellas Pharma, and Seattle Genetics/Astellas. Y.C. reports stock ownership in Oric Pharmaceuticals and sponsored Research from Foghorn Therapeutics. J.E.R is a Consultant for Seagen, Astellas, Bayer, AstraZeneca, QED Therapeutics, Merck, Genentech, Infinity, Gilead, Boehringer Ingelheim, Tyra, Mirati, Pfizer, EMD-Serono; reports sponsored research from Seagen, Astellas, Bayer, AstraZeneca, QED Therapeutics, Genentech; and received honoraria from Pfizer and EMD-Serono. The remaining authors declare no conflicts of interest.

## Additional information

Joshua I. Warrick ®[1,18], Wenhuo Hu ®[2,18], Hironobu Yamashita[1,3,18], Vonn Walter ®[4], Lauren Shuman ®[1,3], Jenna M. Craig[1,3], Lan L. Gellert[5], Mauro A. A. Castro[6], A. Gordon Robertson[7], Fengshen Kuo ®[8], Irina Ostrovnaya[9], Judy Sarungbam[10], Ying-bei Chen ®[10], Anuradha Gopalan[10], Sahussapont J. Sirintrapun[10], Samson W. Fine ®[10], Satish K. Tickoo ®[10], Kwanghee Kim ®[8], Jasmine Thomas[8], Nagar Karan[8], Sizhi Paul Gao[2], Timothy N. Clinton ®[8], Andrew T. Lenis ®[8], Timothy A. Chan ®[2], Ziyu Chen ®[2], Manisha Rao[2], Travis J. Hollman ®[9], Yanyun Li[9], Nicholas D. Socci[11], Shweta Chavan[11], Agnes Viale[11], Neeman Mohibullah[11], Bernard H. Bochner[8], Eugene J. Pietzak[8], Min Yuen Teo ®[12], Gopa Iyer ®[12], Jonathan E. Rosenberg[12], Dean F. Bajorin ®[12], Matthew Kaag[3], Suzanne B. Merrill[3], Monika Joshi[13], Rosalyn Adam ®[14], John A. Taylor III[15], Peter E. Clark[16], Jay D. Raman[3], Victor E. Reuter[9], Yu Chen ®[2,11], Samuel A. Funt ®[11], David B. Solit ®[2,11,12], David J. DeGraff ®[1,3,17] ✉ & Hikmat A. Al-Ahmadie ®[9] ✉

[1]Department of Pathology and Laboratory Medicine, Pennsylvania State University College of Medicine, Hershey, PA, USA. [2]Human Oncology and Pathogenesis Program, Memorial Sloan Kettering Cancer Center, New York, NY, USA. [3]Department of Urology, Pennsylvania State University College of Medicine, Hershey, PA, USA. [4]Department of Public Health Sciences, Pennsylvania State University College of Medicine, Hershey, PA, USA. [5]Department of Pathology, Microbiology, and Immunology, Vanderbilt University Medical Center, Nashville, TN, USA. [6]Bioinformatics and Systems Biology Laboratory, Federal University of Parana, Curitiba, Paraná, Brazil. [7]BC Cancer, Canada's Michael Smith Genome Sciences Centre, Vancouver, BC, Canada. [8]Urology Service, Department of Surgery, Memorial Sloan Kettering Cancer Center, New York, NY, USA. [9]Department of Epidemiology and Biostatistics, Memorial Sloan Kettering Cancer Center, New York, NY, USA. [10]Department of Pathology and Laboratory Medicine, Memorial Sloan Kettering Cancer Center, New York, NY, USA. [11]Marie-Josée and Henry R. Kravis Center for Molecular Oncology, Memorial Sloan Kettering Cancer Center, New York, NY, USA. [12]Department of Medicine, Memorial Sloan Kettering Cancer Center, New York, NY, USA. [13]Department of Medicine, Division of Hematology-Oncology, Penn State Cancer Institute, Hershey, PA, USA. [14]Department of Urology, Boston Children's Hospital, Boston, MA, USA. [15]Department of Urology, University of Kansas Medical Center, Kansas City, MO, USA. [16]Levine Cancer Institute, Atrium Health, Charlotte, NC, USA. [17]Deparment of Biochemistry and Molecular Biology, Pennsylvania State University College of Medicine, Hershey, PA, USA. [18]These authors contributed equally: Joshua I. Warrick, Wenhuo Hu, Hironobu Yamashita. ✉e-mail: ddegraff@pennstatehealth.psu.edu; alahmadh@mskcc.org

