## [Peer Review File · Nature Communications]

FOXA1 repression drives lineage plasticity and immune heterogeneity in bladder cancers with squamous differentiationReviewers' Comments:

Reviewer #1:

Remarks to the Author:

In this paper, the authors performed a series of genomic analyses of mixed histology bladder cancers with separable regions of urothelial and squamous differentiation, intending to define the morphologic heterogeneity in urothelial carcinoma. Based on the analytical results, they concluded that lineage plasticity with squamous differentiation is a marker for intratumoral genomic and immunologic heterogeneity in patients with bladder cancer and a biomarker of intrinsic immunotherapy resistance. Specifically, the authors carried out a scRNA-seq data analysis, aiming to figure out the expression differences in squamous morphology states. However, the analysis of scRNA-seq data is shallow and cannot be reproduced, based on the limited information provided. Detailed major comments regarding scRNA-seq analysis are listed below.

1. The author did not provide anything specific regarding the generated scRNA-seq data. The raw sequencing data or expression matrix is not publicly available; descriptions to library preparation are vague; details of analysis and specific parameters used in Seurat and Slingshot were missing; statistical results were missing. It is impossible to evaluate and reproduce the result of scRNA-seq data with the current information provided in the manuscript.
2. As listed in Table S1, the source of tissue for scRNA-seq data is n/a, what does that mean? Also, it is unclear which cells were collected from the SqD and UC region, which makes me confused about the conclusion "single cell transcriptomic analyses also provide molecular evidence that such a transition is well underway in regions of UC prior to discernable morphologic evidence of SqD at the histologic level".
3. It is unclear how many cells were detected in the raw single-cell data and how those cells were distributed. How were immune and stromal cell populations determined by the three genes in Figure S5? Any expression thresholds? Based on Figure S5B-D, all cells seemed to be either immune cells or epithelial cells. It is pointless to perform Seurat cell clustering if such cell filtering is only based on the expression level of some marker genes.
4. For Figure S5A-D, tSNE is a method for dimensional reduction, not for cell clustering, and it should be UMAP rather than tSNE used here. Seurat applied PCA dimension reduction followed by the Louvain clustering method for cell clustering. tSNE/UMAP is only to visualize and explore the result. For Figure S5E, it should be differentially expressed gene (DEG) analysis, not cluster analysis.
5. I am wondering about the matchiness between Seurat and Slingshot cell clustering results, and any comparisons and discussions regarding the similarity or differences between the DEGs identified from the two results are preferred (Figure S5E and S5A).
6. In Line 173-175, the authors claimed that the GSEA results revealed that some genes play a role in normal epidermis development, which are consistent with the scRNA-seq data analysis. What are those genes? Statistically, consistent means 100% match or significantly similar (needs a p-value), is it such a case here?
7. The deconvolution analysis in this paper was using CIBERSORT, which is outdated. The current advanced methods include CIBERSORT+, TIMER 2.0, etc. The authors should clearly justify the use of CIBERSORT, or directly apply the new version.

Reviewer #2:

Remarks to the Author:

The manuscript, submitted by Warrick et al., reports that squamous basal-like bladder carcinoma shares a number of genomic alterations with urothelial luminal-like carcinoma; albeit the former histotype exhibits significant differences in the immune landscape and, importantly, is refractory to PDL-1-based immunotherapy. The authors have shown that one of the key drivers, FOXA1, ensures the maintenance of the luminal-like histotype. FOXA1 expression is lost in squamous tumors and is associated with the expression of PDL-1. The transition from luminal to squamous phenotype is gradual, with gene signatures of squamous differentiation already detected in the luminal histotype

prior to the acquisition of the squamous phenotype. This differentiation program bears resemblance to the mode of differentiation in the epidermis: from basal to supra-basal and to cornified layers.

Criticisms

The coexistence of the two histotypes seems to represent only 10% of the cohort of bladder cancers used in this study. Moreover, only 21 samples were able to be macrodissected. The authors do not discuss the potential presence of other coexisting histotypes. The micrograph presented in Figure 1A should be improved. The luminal-like tumor is highly infiltrated by lymphocytes. Is this a common feature in mixed tumors?

Figure 2A reports that 8 out of 12 luminal-like tumors exhibited expression profiling of basal squamous-like tumors even if in Figure 2B, one can observe some shift using the BACE47 and UPK/KRT signatures. This raises an important issue regarding the role of FOXA1 in the maintenance of the luminal-like phenotype and the gradual trans-differentiation into a squamous phenotype.

Figure 2C shows single-cell sequencing of a tumor predominantly harboring a squamous phenotype. This analysis does not allow us to visualize the progressive trans-differentiation occurring in the luminal-like tumors, or allow us to determine at which stage FOXA1 is downregulated.

Figure 3A shows that one squamous tumor harbors a well-defined luminal phenotype, as assessed by its differential gene expression. Have the authors reviewed the histopathology of this tumor?

Figure 3D shows that FOXA1 is poorly expressed in the luminal-like tumors. Yet, GATA3 and PPRG exhibit much higher expression levels but they are not analyzed in this study.

The immune signatures of most luminal and squamous tumors do not reveal significant differences suggesting an increased immune refractoriness of the squamous tumors. The presence of an IFN- γ signature and M2 macrophages in the luminal tumors may indicate that these tumors are also refractory to immunotherapy (Figure 4).

Figure 4C does not show what is described in the Results section. What is the meaning of the *? Is Figure 4C right panel legend meant to say "SpD vs UC" rather than the reverse?

In Figure 4E, the PDL-1 histochemistry is not convincing. PDL-1 is mostly expressed in blood vessels and possibly in stroma. In case #2, the right panel does not exhibit a typical squamous phenotype.

Figure S9 is extremely hard to interpret.

In Figure 5G, the survival curves do not show a significant survival benefit for FOXA1-positive tumors.

In Figure 5H, it is difficult to interpret whether FOXA1 plays a key role in a number of tumor types aside from bladder carcinoma; where would be the cutoff for correlation?

Figure 6B is poorly described. Is the analysis duplicated?

The figure legend for Figure 6C needs clarification. The left panel shows two clusters with cluster one representing 986 peaks and cluster two representing 113 peaks. However, in the left heatmap, cluster two shows significantly more peaks than cluster one: How do the authors explain this? How have these clusters have been defined? The difference between the two clusters is unclear. The figure legend pertaining to the right panel is also unclear. Why are H3K27ac promoter regions associated with downregulation of gene expression?

Figure 6F does not show the ChIP-seq peak value.

This study did not show FOXA1 ChIP-seq data to assess FOXA1 binding on these enhancer/promoter regions. It would also be necessary to investigate how FOXA1 could regulate its proximal promoter activity. Additionally, a mechanism describing how FOXA1 regulates PD-L1 expression is missing.

Reviewer #3:

Remarks to the Author:

Lineage plasticity and immune heterogeneity are coordinately dysregulated by FOXA1 repression in bladder cancers with squamous differentiation

In this manuscript by Warrick I.J. et.al., the authors study the genomics of the bladder cancers to uncover clinically relevant findings, especially in those carcinomas that exhibit morphological heterogeneity. They found out that absence of FOXA1 causes increased PDL1 expression, which could

be a possible mechanism as to why anti- PDL1 immunotherapy may not be effective in bladder tumors of basal subtype with squamous differentiation. This also underscores how gene expression analysis can help in addressing certain current limitations in classification of mixed morphology tumors. The findings are novel and comprehensive analysis has been carried out. Overall, this study has relevance in in diagnosis and treatment.

A few points that would benefit this study are

1. In Page 10, the authors state 'The ability of FOXA1 to regulate chromatin accessibility at enhancer and promoter elements via displacement of linker histones is a hallmark characteristic of this and other pioneer transcription factors [48-50].'

To fully support this statement, the loss of linker histones at the site of PDL-1 enhancers needs to be shown experimentally, in addition to citing references.

2. In Page 12 the authors state 'While WES confirmed that both the UC and SqD regions of individual tumors were derived from a shared precursor cell, we observed significant mutational discordance between the two morphologically distinct regions consistent with early branched evolution.'

The possibility of multiclonal origin of cancer cannot be ruled out. On what basis is it certain that the cells have a shared precursor?

Minor edits

1. Phylogeny plots: In Figure S3B, it is mentioned 'Shared mutations between UC and SqD were indicated with dark blue on the top left of each panel.' Is it dark blue or green, since in Figure 1D and 1E, it is mentioned as green; and blue, in fact is referenced – 'the cancer cell fraction (CCF) of individual mutations is indicated by the degree of blue shading.

It would be helpful to incorporate a section in the Methods about phylogeny.

2. In Figure 6C, what is the basis of classification as Cluster 1 and Cluster 2?

Reviewer #1, expert in scRNA-seq and WES (Remarks to the Author):

In this paper, the authors performed a series of genomic analyses of mixed histology bladder cancers with separable regions of urothelial and squamous differentiation, intending to define the morphologic heterogeneity in urothelial carcinoma. Based on the analytical results, they concluded that lineage plasticity with squamous differentiation is a marker for intratumoral genomic and immunologic heterogeneity in patients with bladder cancer and a biomarker of intrinsic immunotherapy resistance. Specifically, the authors carried out a scRNA-seq data analysis, aiming to figure out the expression differences in squamous morphology states. However, the analysis of scRNA-seq data is shallow and cannot be reproduced, based on the limited information provided. Detailed major comments regarding scRNA-seq analysis are listed below.

1. The author did not provide anything specific regarding the generated scRNA-seq data. The raw sequencing data or expression matrix is not publicly available; descriptions to library preparation are vague; details of analysis and specific parameters used in Seurat and Slingshot were missing; statistical results were missing. It is impossible to evaluate and reproduce the result of scRNA-seq data with the current information provided in the manuscript.

We thank the reviewer for this comment. We have now provided more details about library preparation, Seurat and Slingshot analysis and statistical results. We have also made the raw sequencing data available to the reviewers for further analysis. The GSE172433 dataset now includes the scRNA-Seq raw data, in addition to the previously uploaded RNA-Seq and ChIP-Seq data generated from the UM-UC-1 isogenic cells. These data can be accessed using access token "kjuhcamejfwpfeb".

The following three paragraphs replace the last paragraph of the methods related to the scRNA-Seq section (lines 448-469):

Single-cell RNA sequencing. Tumor tissue was processed immediately after removal from the bladder. Tumor tissue was cut into small pieces (<1 mm in

greatest dimension) and incubated in 1 ml media supplied with enzymes from the tumor dissociation kit (Miltenyi, Cat: 130-095-929) for 30 minutes on a 37 °C shaker. Subsequently, 10 ml DMEM media was added to dilute the suspension, then a 40- μ m cell mesh was used to filter the suspension. After centrifugation at 250g for 5 min, the supernatant was discarded, and the cells were washed with DMEM media twice. The pellet was resuspended in 1 ml of calcium- and magnesium-free PBS containing 0.04% weight/volume BSA. Finally, 10 μ l of suspension was counted under an inverted microscope with a hemocytometer. Trypan blue was used to quantify live cells.

Following the manufacturer's protocol, the Chromium Single cell 3' Reagent v3 kit was used to prepare barcoded scRNA-seq libraries. Single-cell suspensions were loaded onto a Chromium Single-Cell Controller Instrument (10x Genomics) to generate single-cell gel beads in emulsions (GEMs). To capture 10,000 cells per library, approximately 15,000 cells were added to each channel. After generation of GEMs, reverse transcription reactions were engaged to generate barcoded full-length cDNA. Next, cDNA was amplified, fragmented, end-repaired, A-tailed, and ligated to an index adaptor, and then the library was amplified. The library was sequenced on a HiSeq X Ten platform (Illumina) to generate 150 bp paired-end reads.

Sequencing reads were aligned to GRCh38 by Cell Ranger (10x genomics, v3.0) and analyzed with the Seurat package v4.0 (Stuart et al. 2019). Cells with UMI numbers <1000 or with over 10% mitochondrial-derived UMI counts were considered low-quality cells and were removed. Doublets were predicted using DoubletFinder, v2.0 and excluded after which 4768 cells remained for downstream analysis. Data normalization, dimension reduction (RunUMAP), and cluster identification (FindClusters) were performed with default parameters. To map differentiation of the tumor cell subpopulations, pseudotime analysis was performed using the Slingshot package in R (Street et al. 2018) with default parameters.

2. As listed in Table S1, the source of tissue for scRNA-seq data is n/a, what does that mean? Also, it is unclear which cells were collected from the SqD and UC region, which makes me confused about the conclusion “single cell transcriptomic analyses also provide molecular evidence that such a transition is well underway in regions of UC prior to discernable morphologic evidence of SqD at the histologic level”.

We thank the reviewer for this comment and apologize for this omission. The tissue source for the scRNA-seq sample was a bladder tumor consisting of urothelial carcinoma with extensive squamous differentiation, which we have now corrected in Table S1. Due to the need for fresh tissue that requires immediate processing for scRNA-seq, we chose an area of the tumor that was most viable by visual inspection, and which we also microscopically confirmed that it is from a mixed histology tumor containing urothelial and squamous areas.

3. It is unclear how many cells were detected in the raw single-cell data and how those cells were distributed. How were immune and stromal cell populations determined by the three genes in Figure S5? Any expression thresholds? Based on Figure S5B-D, all cells seemed to be either immune cells or epithelial cells. It is pointless to perform Seurat cell clustering if such cell filtering is only based on the expression level of some marker genes.

As above, there were 4768 live cells that were analyzed based on the cut-off of mitochondrial gene expression of <15% and number of genes detected of >500 per cell. Cell subtypes were identified using the SingleR package which revealed B cells (23 cells), myeloid-derived suppressor cells-MDSC (667 cells), dendritic cells-DC (14 cells), monocyte (682 cells), NK cells (1398 cells), T cells (1133 cells), smooth muscle cells (23 cells), and epithelial/tumor cells (728 cells). This more detailed information is now provided in the modified Figure S5.

4. For Figure S5A-D, tSNE is a method for dimensional reduction, not for cell clustering, and it should be UMAP rather than tSNE used here. Seurat applied PCA dimension reduction followed by the Louvain clustering method for cell clustering. tSNA/UMAP is only to visualize and explore the result.

For Figure S5E, it should be differentially expressed gene (DEG) analysis, not cluster analysis.

We thank the reviewer for this comment. We tried both tSNE and UMAP analysis in our study, and the final results in this manuscript are from the UMAP analysis as suggested by the reviewer. We have now corrected the label in the figure legend from tSNE to UMAP. Additionally, Figure S5 has been revised with additional cell types identified using SingleR packages: Violin plots of PTPRC (CD45), KRT17, and KRT6A gene expression, and a heatmap based on the top 50 up-regulated genes for each cluster identified by Seurat packages. Figure S5 legend has been revised to the following:

Figure S5. Single Cell RNA-Seq (scRNA-Seq) from a bladder urothelial carcinoma with extensive SqD. A) Sub-populations were identified by applying dimension reduction (UMAP) and cluster analysis. B) Cell types identified by SingleR package using reference gene expression from Human Primary Cell Atlas database. C) Expression of PTPRC, KRT17, and KRT6A among the sub-populations identified. D) Cluster analysis of top 50 up-regulated genes within each sub-population.

5. I am wondering about the matchiness between Seurat and Slingshot cell clustering results, and any comparisons and discussions regarding the similarity or differences between the DEGs identified from the two results are preferred (Figure S5E and S5A).

We thank the reviewer for this comment. We plotted the Slingshot analysis with both pseudotime or Seurat cluster and compared with the cluster analysis by Seurat package. As show below, the clusters from Seurat cluster analysis as in our manuscript (right panel) are consistent with the cell positions from Slingshot analysis (left and middle panels), even though we did notice that there was a slight difference in cell distribution between the Slingshot and Seurat clustering.

6. In Line 173-175, the author s claime d that the GSEA results reveal

ed that some genes play a role in normal epidermis development, which are consistent with the scRNA-seq data analysis. What are those genes? Statistically, consistent means 100% match or significantly similar (needs a p-value), is it such a case here?

As suggested, we have now listed the gene names that significantly changed expression ($FDR < 0.05$) in the heatmap of Figure S7 including the GO Epidermis development pathway, GO Cornified development pathway, and GO epithelial cell differentiation pathway. The Figure S7 legend has also been updates to the following:

Figure S7. Bulk RNA-Seq analysis showing significantly differentially expressed genes ($FDR < 0.05$) in 12 paired UC and SqD areas from bladder tumors. Cluster analysis genes from (A) GO epidermis development gene set, (B) cornified envelope gene set, and (C) GO epithelial cell differentiation. D) Changes in RNA expression levels of the 3 transcription factors FOXA1, GATA3 and PPARG between UC and SqD regions.

7. The deconvolution analysis in this paper was using CIBERSORT, which is outdated. The current advanced methods include CIBERSORT+, TIMER 2.0, etc. The authors should clearly justify the use of CIBERSORT, or directly apply the new version.

At the reviewer's suggestion, we repeated our analysis using CIBERSORTX and the results were overall similar to those of the original analysis with

CIBERSORT. In general, SqD areas tended to have lower immune infiltration, which was a main conclusion of our findings and discussion. In the revised manuscript, we have included the results from CIBERSORTX that replaced the original figure 4C and included additional analysis by EPIC in Figure S8 that further supports our findings.

Reviewer #2, expert in bladder cancer subtypes (Remarks to the Author) The manuscript, submitted by Warrick et al., reports that squamous basal-like bladder carcinoma shares a number of genomic alterations with urothelial luminal-like carcinoma; albeit the former histotype exhibits significant differences in the immune landscape and, importantly, is refractory to PDL-1-based immunotherapy. The authors have shown that one of the key drivers, FOXA1, ensures the maintenance of the luminal-like histotype. FOXA1 expression is lost in squamous tumors and is associated with the expression of PDL-1. The transition from luminal to squamous phenotype is gradual, with gene signatures of squamous differentiation already detected in the luminal histotype prior to the acquisition of the squamous phenotype. This differentiation program bears resemblance to the mode of differentiation in the epidermis: from basal to supra-basal and to cornified layers.

Criticisms

The coexistence of the two histotypes seems to represent only 10% of the cohort of bladder cancers used in this study. Moreover, only 21 samples were able to be macrodissected. The authors do not discuss the potential presence of other coexisting histotypes. The micrograph presented in Figure 1A should be improved. The luminal-like tumor is highly infiltrated by lymphocytes. Is this a common feature in mixed tumors?

We thank the reviewer for this comment. Squamous differentiation is detected in approximately 30% of invasive urothelial carcinomas, making it the most common divergent differentiation observed in bladder cancers. Being present in 10% of the MSK-IMPACT cohort reflects the type of tumors that underwent next generation sequencing at our institution during this period and suggests that tumors with squamous differentiation might have been relatively underrepresented in our dataset. Nonetheless, and despite that, we have identified 82 tumors that exhibited squamous differentiation that formed the basis for this comparison between the genomics of UC NOS (n=587) and those of UC with SqD (n=82).

Regarding the broader whole exome analysis, the 21 samples analyzed were chosen as they had distinct regions of urothelial and squamous morphology that could be readily separated/macrodissected to allow for genomic analysis. In many bladder tumors, areas of urothelial and squamous morphology are intermixed, and it is thus difficult to separate the UC and SqD regions for this type of analysis. Single cell RNA sequencing may help in overcoming such challenges that are inherent in intermixed tumors as we have shown in the example that we included in the paper. Finally, the 21 tumors analyzed in this report did not contain

histotypes other than urothelial and squamous. We clarified the latter point in the patient samples section of the Material and Methods.

At the reviewer's request, we have also modified Figure 1A to include a photomicrograph from another field of the urothelial (luminal-like) area. We would also like to point out that the original photomicrograph was taken from an area rich with tumor, not immune cells, and the "lymphocytes" pointed out by the reviewer are infiltrating tumor cells invading muscle tissue of the bladder wall. We hope that the new photomicrograph is clearer and sufficient to remove any ambiguity about the nature and composition of the tissue sequenced.

Figure 2A reports that 8 out of 12 luminal-like tumors exhibited expression profiling of basal squamous-like tumors even if in Figure 2B, one can observe some shift using the BASE47 and UPK/KRT signatures. This raises an important issue regarding the role of FOXA1 in the maintenance of the luminal-like phenotype and the gradual trans-differentiation into a squamous phenotype.

We thank the reviewer for this comment. We fully agree with this statement.

Figure 2C shows single-cell sequencing of a tumor predominantly harboring a squamous phenotype. This analysis does not allow us to visualize the progressive trans-differentiation occurring in the luminal-like tumors, or allow us to determine at which stage FOXA1 is downregulated.

We totally agree with and thank the reviewer for this insightful comment. This, unfortunately, is an inherent limitation of any single-cell RNAseq analysis as the type of tissue undergoing analysis cannot be determined beforehand, due primarily to the absolute need for both fresh and viable tissue for this analysis which requires urgency in obtaining a tissue aliquot and initiating the analysis without delay. After ample sampling of this tumor, which was done from FFPE tissue, and obviously following the performance of sc-RNAseq, the tumor was determined to be of extensive squamous differentiation, which would explain the only small number of cells expressing FOXA1 in Figure 2E. Therefore, this tumor appears to be at the extreme end of squamous/basal differentiation spectrum. An ideal sample would have been a tumor fragment that contained a more even mixture of urothelial and squamous cells that would have allowed for the identification of the timing of FOXA1 downregulation. In the future, we hope to preferentially target for scRNA-seq tumors with more morphologic heterogeneity and divergent differentiation to help us understand better the biology of trans-differentiation between different morphologic and molecular subtypes.

Figure 3A shows that one squamous tumor harbors a well-defined luminal phenotype, as assessed by its differential gene expression. Have the authors reviewed the histopathology of this tumor?

We thank the reviewer for this comment. We can confirm that the tumor in question does have squamous morphology but not as extensive as in other SqD areas of other tumors, which is further supported by the fact that it clustered adjacent to the UC area of the same tumor (the updated Figure 3A shows that the UC and SqD areas of tumor #11 are next to each other). This heatmap was generated by cluster analysis following the identification of differentially expressed genes between the UC and SqD regions. This sample does not show the same level of differential expression of the genes that were used to generate the heatmap. Of note, we also confirm that the expression of the 3 transcription factors FOXA1, GATA3, PPARG is lower in the SqD region of the sample than in the UC region of this tumor as shown below.

Figure 3D shows that FOXA1 is poorly expressed in the luminal-like tumors. Yet, GATA3 and PPRG exhibit much higher expression levels but they are not analyzed in this study.

We agree with the reviewer's observation that the examples included in this figure show that the level of FOXA1 expression is not as strong as those of GATA3 and PPAR α . Out of concern that the image of FOXA1 IHC was perhaps not representative, we replaced the FOXA1 IHC example with another field of the same tumor that shows a stronger intensity of stain, albeit not as robust as the other two markers. But this suggestion by the reviewer and the new example of IHC also highlights an important point - that expression as shown by IHC is not always a reflection of the biologic role of genes. The choice to study FOXA1 was based on its role as a pioneer factor that is important in the development and maintenance of urothelial/luminal phenotype. We agree that more detailed studies of the *GATA3* and *PPARG* are warranted as part of future studies.

The immune signatures of most luminal and squamous tumors do not reveal significant differences suggesting an increased immune refractoriness of the squamous tumors. The presence of an IFN- signature and M2 macrophages in the luminal tumors may indicate that these tumors are also refractory to immunotherapy (Figure 4).

We fully agree with the reviewer's observation. The main point that we wanted to stress is the intratumoral immune heterogeneity rather than any specific immune signature. Despite the presence of a IFN γ immune signature in the overwhelming majority of samples (both UC and SqD), intratumoral immune heterogeneity was evident between UC and SqD regions of individual tumors in all cases.

Figure 4C does not show what is described in the Results section. What is the meaning of the *? Is Figure 4C right panel legend meant to say "SpD vs UC" rather than the reverse?

Convention

We apologize for this oversight and have corrected the discrepancy between the figure and the results. The * indicates significant difference in expression of markers between UC and SqD which is now highlighted in the figure legend.

In Figure 4E, the PDL-1 histochemistry is not convincing. PDL-1 is mostly expressed in blood vessels and possibly in stroma. In case #2, the right panel does not exhibit a typical squamous phenotype.

We can confirm to the reviewer and the editorial team that PD-L1 expression in original figure 4E is on tumor cells that is accentuated at the periphery near the tumor-stromal interface. To remove any ambiguity, we replaced images corresponding to Case #1 with new images taken from different areas of the tumor at higher magnification to highlight the differences in PD-L1 expression on tumor cells between UC and SqD areas.

For Case #2, the choice to take the photomicrograph at this low magnification was meant to highlight the contrast of PD-L1 expression in the different regions of the tumor that would not have been captured at higher magnification. To remove any ambiguity, we have now provided higher magnification images from SqD regions in a new supplementary figure (Figure S13) in which multiple examples of unequivocal squamous differentiation are depicted.

Figure S9 is extremely hard to interpret.

We apologize for the difficulty interpreting this figure. We have expanded the figure legend to better explain the findings in this figure. The new figure legend reads:

Figure S9. Multicolor immunofluorescence staining for immune and tumor cells in UC (left) and SqD (right) regions from the same tumor. There is PD-L1 overexpression on tumor cells in SqD compared to UC regions (note tumor cells co-expressing p63, as white nuclear marker, and PD-L1, as red membranous and cytoplasmic marker). Additionally, there is variable expression and distribution of CD4+ and CD8+ T-cells and macrophages (CYP27A1) in SqD and UC regions.

In Figure 5G, the survival curves do not show a significant survival benefit for FOXA1-positive tumors.

We thank the reviewer for this comment. Regulon analysis is an approach utilizing mutual information for analysis of expression data. This is a powerful and accepted approach for inferring transcription factor activity within samples. To clarify, we included a statement to this effect on page 10, the first time this approach is introduced. Here, we performed regulon analysis on a publicly available databases from two published studies (the bladder TCGA cohort [ref #13] and Sjordahl et al. [ref # 49]) and NOT our own cohort. This analysis showed that, in the bladder TCGA cohort, samples with low FOXA1 regulon activity (Figure 5F and 5G; blue) are enriched in CD274/PD-L1 expression and the basal-squamous molecular subtype, and are associated with significantly worse disease-specific survival ($p=0.01$).

In Figure 5H, it is difficult to interpret whether FOXA1 plays a key role in a number of tumor types aside from bladder carcinoma, where would be the cutoff for correlation?

We thank the reviewer for this comment. The point that we are trying to convey from this figure is the correlation between FOXA1 and CD274 mRNA expression levels across the many cancer subtypes that were analyzed by the TCGA. Our results show that there is an inverse relationship between FOXA1 and PD-L1 mRNA levels in only a few cancer types, but the most pronounced inverse correlation exists in bladder cancer. These data suggest that the association between FOXA1 and PD-L1 is not tumor agnostic. For clarification, we added the following paragraph to the methods section to better describe this analysis:

Gene expression correlation analysis. Gene expression data of each cancer cohort in the Cancer Genome Atlas (TCGA) project were analyzed by TCGAAbiolinks (Bioconductor package). The expression correlations between CD274 and FOXA1 were calculated based on the normalized reads using `cor.test` function in R with Spearman method. The results of correlation factors and p values were summarized and FDR adjusted p values were calculated by `p.adjust` function with Bonferroni correction.

Figure 6B is poorly described. Is the analysis duplicated?

We thank the reviewer for this comment. The analysis was performed in duplicates. We have modified Figure 6 and updated the legend accordingly.

The figure legend for Figure 6C needs clarification. The left panel shows two clusters with cluster one representing 986 peaks and cluster two representing 113 peaks. However, in the left heatmap, cluster two shows significantly more peaks than cluster one: How do the authors explain this? How have these clusters have been defined? The difference between the two clusters is unclear. The figure legend

pertaining to the right panel is also unclear. Why are H3K27ac promoter regions associated with downregulation of gene expression?

To address this comment, we now provide an updated Figure 6 that includes analysis of FOXA1 ChIP-seq data. An updated figure legend is also provided that explains the findings.

The two clusters were identified by deeptools with kmean = 2.

Cluster analysis is based on H3K27ac-modified enhancer regions enriched in either parental or FOXA1 KO UMUC-1 cells, FOXA1 binding coverage around the same region, and the expression of genes associated with these sites.

Figure 6F does not show the ChIP-seq peak value.

We thank the reviewer for this comment. We now added the peak scale for H3K27ac (0-80) and for FOXA1 (0-20).

This study did not show FOXA1 ChIP-seq data to assess FOXA1 binding on these enhancer/promoter regions. It would also be necessary to investigate how FOXA1 could regulate its proximal promoter activity. Additionally, a mechanism describing how FOXA1 regulates PD-L1 expression is missing.

Based on this suggestion, we have performed an analysis of FOXA1 ChIP-seq using the UM-UC-1 parental cells, which gave us an opportunity to perform an integrated analysis of H3K27ac changes in gene expression within the context of identified FOXA1 binding sites. These new results have been added to the updated Figure 6 and new Figure S12. These results show that: (1) both increases and decreases in H3K27ac are associated with increased and decreased gene expression, respectively; and (2) increases and decreases in H3K27ac occur in regions bound by FOXA1 in parental cells. While we cannot exclude additional potential mechanisms contributing to the regulation of PD-L1 by FOXA1, we do identify several FOXA1 binding sites within cis-regulatory regions of the CD274 gene (also shown in updated Figure 6). Currently, our hypothesis is that FOXA1 recruits (as yet unknown) HDACs (to inhibit CD274 acetylation and expression), and that FOXA1 inactivation relieves this repressive function. However, we feel that testing this hypothesis is beyond the scope of the current work, and we look forward to addressing this possibility in context of future studies.

Reviewer #3, expert in ChIP-seq and RNA-seq (Remarks to the Author):

Lineage plasticity and immune heterogeneity are coordinately dysregulated by FOXA1 repression in bladder cancers with squamous differentiation

In this manuscript by Warrick I.J. et.al., the authors study the genomics of the bladder cancers to uncover clinically relevant findings, especially in those

carcinomas that exhibit morphological heterogeneity. They found out that absence of FOXA1 causes increased PDL1 expression, which could be a possible mechanism as to why anti- PDL1 immunotherapy may not be effective in bladder tumors of basal subtype with squamous differentiation. This also underscores how gene expression analysis can help in addressing certain current limitations in classification of mixed morphology tumors. The findings are novel and comprehensive analysis has been carried out. Overall, this study has relevance in in diagnosis and treatment.

We thank the reviewer for these encouraging comments on our study and greatly appreciate the overall positive assessment of our manuscript.

A few points that would benefit this study are

1. In Page 10, the authors state ‘The ability of FOXA1 to regulate chromatin accessibility at enhancer and promoter elements via displacement of linker histones is a hallmark characteristic of this and other pioneer transcription factors [48-50].’ To fully support this statement, the loss of linker histones at the site of PDL-1 enhancers needs to be shown experimentally, in addition to citing references.

We agree with the reviewer and have therefore removed the sentence from the introduction to our functional experiment.

2. In Page 12 the authors state ‘While WES confirmed that both the UC and SqD regions of individual tumors were derived from a shared precursor cell, we observed significant mutational discordance between the two morphologically distinct regions consistent with early branched evolution.’ The possibility of multiclonal origin of cancer cannot be ruled out. On what basis is it certain that the cells have a shared precursor?

In response to the reviewer’s comment, we have performed clonality analysis on the paired samples from individual tumors to confirm that the UC and SqD regions of each tumor were clonally related and could not have arisen independently, supporting a shared origin. We have also added the following text and references to the clonality analysis in the main manuscript (lines 108-109):

Clonality analysis further supported that the two components were clonally related and did not arise independently ($p=0$, Table S2) [16, 17].

And added the following text to the materials and methods of whole exome sequencing section (lines 422-430):

For **clonality** analysis, the Clonality R package [16] was applied to test whether a pair of tumors from the same patient are clonal (derived from a common precursor) or independent (distinct primary tumors). When analyzing somatic mutation data, as was done here, the Clonality R package applies a statistical test

developed by Ostrovnaya et al. [17]. In brief, a conditional likelihood model is applied to test the null hypothesis that the tumors are independent. The model only uses loci where at least one of the two tumors has a mutation. Marginal mutation frequencies are estimated from an external reference, and in this analysis the frequencies were derived from mutations observed in the TCGA BLCA cohort [13]. Then a generalized likelihood ratio test is performed based on the number of observed mutations that are common to both tumors or appear in exactly one of the two tumors. The Clonality output is presented in Table S2.

We also added output data of the analysis in Table S2.

Minor edits

- 1. Phylogeny plots: In Figure S3B, it is mentioned ‘Shared mutations between UC and SqD were indicated with dark blue on the top left of each panel.’ Is it dark blue or green, since in Figure 1D and 1E, it is mentioned as green; and blue, in fact is referenced – ‘the cancer cell fraction (CCF) of individual mutations is indicated by the degree of blue shading. It would be helpful to incorporate a section in the Methods about phylogeny.**

We thank the reviewer for pointing this out and apologize for the oversight. The figure legend of Figure S3B has been corrected as per the reviewer’s comment. The following paragraph on phylogeny was also added to the methods under the whole exome sequencing section (lines 431-435):

For **phylogeny** analysis, evolutionary relationships of mutation phylogeny between UC and SqD regions were inferred using the union of somatic mutations called in any of the paired samples, a normal sample with blank of these mutations was added for each patient. Package “Analyses of Phylogenetics and Evolution” (R package “ape”) was used to infer the phylogenetics relationship based on the cluster results of the mutation table of the UC, SqD and the assumed normal sample.

- 2. In Figure 6C, what is the basis of classification as Cluster 1 and Cluster 2?**

To clarify, the clusters were defined by kmean (k=2). We have also now explained the methods and results in greater detail in the modified Figure 6 legend.

We hope that we were able to adequately and satisfactorily address the reviewers’ comments and look forward to your feedback.

Sincerely,

Hikmat Al-Ahmadie and David DeGraff

Reviewers' Comments:

Reviewer #1:

Remarks to the Author:

The authors have answered all my questions and revised them properly. I have no further concerns.

Reviewer #2:

Remarks to the Author:

The response to reviewer 2 is somewhat satisfactory. However the mechanism of transdifferentiation to a squamous histotype remains elusive. As stated by the authors, it would be necessary to analyze more appropriate samples to study lineage tracing during the transdifferentiation process. In addition, the role of Foxa1 remains elusive. Additional data would reinforce the current study and clarify these important issues.

Reviewer #3:

Remarks to the Author:

The results claim that lineage plasticity of squamous differentiation is a marker of intratumoral heterogeneity and biomarker of intrinsic immunotherapy resistance.

It would have been interesting to have the link between FOXA1 and the PD-L1 (CD274) link be explored since it forms an important basis of this study. Merely taking off the statement from the previous version, renders the mechanistic portion weak.

The authors claim that lineage plasticity and genetic heterogeneity can be critical determinants in therapy due to immune signatures being altered, but provide insufficient evidence to show how it occurs.

The scRNA data alone from a few samples cannot justify this conclusion.

Overall,

The clonal origin conclusion is weak.

The authors say that (1) mixed histologies is associated with mixed genetic heterogeneity and altered immune signatures (shown) and (2) therefore will not respond to anti- PDL1 therapies (insufficient data).

(3) FOXA1 and PD-L1 inverse correlation has been reported by others. So the only novelty is that it is found here too in bladder cancers; but hasn't been explored more.

It would be better to replace Chromatin instead of DNA in 'ChIP-seq was performed using DNA extracted..' (Fig 6 legend).

The FOXA1 ChIP-seq Figure 6F needs to be more polished with the CD274 gene TSS prominent and the gray bar being labelled or mentioned in the legend.

The Figure S12 ChIP-seq of H3K27ac: How were enhancers defined as 62% of H3K27ac ChIP-seq? Enhancers can be long range too.

FOXA1 cannot be said to be a repressor of PD-L1 without providing any data to support this.

RE: Response to reviewers (NCOMMS-21-13347A, Lineage plasticity and immune heterogeneity are coordinately dysregulated by FOXA1 repression in bladder cancers with squamous differentiation)

Dear Reviewers,

We appreciate your time and efforts and thank you for your comments. Below is a point by point response to the Reviewers' comments. All newly added text in the manuscript, figure legends and references, is highlighted in yellow.

Reviewers' comments:

Reviewer #1, expert in scRNA-seq and WES (Remarks to the Author):

The authors have answered all my questions and revised them properly. I have no further concerns.

- We thank the Reviewer for this comment and are glad that we answered all the Reviewer's questions.

Reviewer #2, expert in bladder cancer subtypes (Remarks to the Author):

The response to reviewer 2 is somewhat satisfactory. However the mechanism of transdifferentiation to a squamous histotype remains elusive. As stated by the authors, it would be necessary to analyze more appropriate samples to study lineage tracing during the transdifferentiation process. In addition, the role of Foxa1 remains elusive. Additional data would reinforce the current study and clarify these important issues.

- We thank the reviewer for this comment. We recognize that the mechanisms underlying the development of squamous differentiation are likely complex and multifactorial and that analysis of additional samples and lineage tracing studies in animal models will likely yield additional insights. Our work suggests that loss of function of FOXA1, GATA3 and PPARG are likely early steps in the development of the squamous phenotype that likely cooperate with other genetic and epigenetic changes to influence the diverse phenotypes that characterize bladder tumors. In sum, we believe that our finding will likely serve as the basis for additional studies by our laboratories and others.

Regarding the role of FOXA1, our additional experiments have shown that FOXA1 suppression results in increased levels of IRF1 and its binding to the CD274 promoter, which in turn upregulates PD-L1 and other interferon response genes independent of INF γ stimulation. We have added these new findings in a

new paragraph to the Results section starting from line 303 to line 339. Two new figures were also added (Figure 7 & Figure S14) to illustrate the new results. The corresponding methods were also added starting from line 587 to line 607. In addition, the Discussion section has been updated to incorporate the new findings (lines 391-404). Three new references have also been added to support our discussion (new references #48, 63 and 77).

Reviewer #3, expert in ChIP-seq and RNA-seq (Remarks to the Author):

The results claim that lineage plasticity of squamous differentiation is a marker of intratumoral heterogeneity and biomarker of intrinsic immunotherapy resistance.

It would have been interesting to have the link between FOXA1 and the PD-L1 (CD274) link be explored since it forms an important basis of this study. Merely taking off the statement from the previous version, renders the mechanistic portion weak.

In response to the reviewer's comment, we have performed additional experiments to investigate the role of FOXA1 in regulating PD-L1 expression. As mentioned in response to Reviewer #2 above, our results revealed that FOXA1 ablation results in significant upregulation of IRF1 and increased binding to the CD274 promoter, which in turn causes upregulation of PD-L1 and other INF γ response genes in a manner that is independent of INF γ stimulation. We now include these new findings in the Results section with more experimental details (lines 303-339) and added new figures (Figure 7 and Figure S14) that summarize the new findings. Additionally, we updated the Discussion section (lines 391-404) and methods section (line 587-607). We also added 3 new references to support our discussion (new references #48, 63 and 77).

The authors claim that lineage plasticity and genetic heterogeneity can be critical determinants in therapy due to immune signatures being altered, but provide insufficient evidence to show how it occurs.

The mechanisms by which immune checkpoint inhibitors exert their antitumor activities are not completely understood and likely multifactorial. Factors that have been consistently shown to correlate with response to anti-PD-1/PD-L1 antibodies include tumor mutation burden, tumor neoantigen loads and PD-L1 expression. Our work shows that all of these parameters are heterogeneous between the two morphologically distinct and macrodissected components of mixed urothelial and squamous bladder tumors and suggest that this intratumoral heterogeneity can give rise to tumor clones that have features associated with immune checkpoint inhibitor resistance. We were also able to support our observation through detailed histomorphologic evaluation of tumor samples from an anti-PD-L1 trial cohort finding significant enrichment within the non-responder group of tumors with morphologic heterogeneity and variant histology (Figure 4A).

The scRNA data alone from a few samples cannot justify this conclusion.

We would like to point out that we did not utilize the scRNAseq sample to suggest correlation with response to immunotherapy. It is not clear to us how the reviewer arrived at this conclusion. Rather, the single cell sequencing data was included to highlight that squamous differentiation is likely a continuous/gradual process that recapitulates the differentiation stages of the normal epidermis, with the squamous morphology representing the extreme of the basal-squamous phenotype.

Overall,

The clonal origin conclusion is weak.

We respectfully disagree with the reviewer's comment about our conclusion on the clonal origin of the urothelial and squamous regions macrodissected from the same tumor. We applied a well-known and well-established method that was developed by one of the authors (Ostrovnyaya) who also developed the statistical tools for characterizing clonal relatedness (a package in R) for public use. As we have reported, our analysis concluded that the urothelial and squamous components are unequivocally clonally related as in all paired samples the two components shared many mutations in addition to their respective private mutations. Moreover, the reviewer does not provide a specific critique to our methodology and does not suggest an alternative method or approach for us to explore.

The authors say that (1) mixed histologies is associated with mixed genetic heterogeneity and altered immune signatures (shown) and (2) therefore will not respond to anti- PDL1 therapies (insufficient data). (3) FOXA1 and PD-L1 inverse correlation has been reported by others. So the only novelty is that it is found here too in bladder cancers; but hasn't been explored more.

We believe that our data support the position that intratumoral morphologic heterogeneity is associated with a lower likelihood of response to immunotherapy as we highlighted earlier in response to another comment by the same reviewer. We have now shown functional mechanisms of interaction between FOXA1 and PD-L1 that we believe is unique in this setting and different from what has been reported in the literature as we highlight above.

It would be better to replace Chromatin instead of DNA in 'ChIP-seq was performed using DNA extracted..' (Fig 6 legend).

As suggested, DNA has been replaced by chromatin (line 716).

The FOXA1 ChIP-seq Figure 6F needs to be more polished with the CD274 gene TSS prominent and the gray bar being labelled or mentioned in the legend.

More details about the CD274 promoter with FOXA1 and IRF1 binding motifs are now shown in more detail in new Figure S14. The grey bars represent unchanged peaks in CD274 enhancer and promoter in parental and FOXA1 KO UM-UC-1 cells. this description has been added to the figure legend (line 732).

The Figure S12 ChIP-seq of H3K27ac: How were enhancers defined as 62% of H3K27ac ChIP-seq? Enhancers can be long range too.

For this analysis, enhancers were defined as 50kb upstream of Transcriptional Start Site (TSS), or 5kb downstream of Transcriptional End Site [TES]). This was defined in the main text of the manuscript (line 268). While we recognize that enhancers can be of longer range, we set this cutoff so as to obtain a general view of the genomic distribution of FOXA1 binding sites. Our results are consistent with prior reports on the distribution of FOXA1 binding sites within the genome (PMID: 18358809 & PMID: 31826955).

FOXA1 cannot be said to be a repressor of PD-L1 without providing any data to support this.

To address this comment by the reviewer, we have now provided functional data showing that FOXA1 binds the *CD274* promoter both in the presence and absence of interferon gamma treatment, indicating that FOXA1 indeed does not directly suppress *CD274* expression. Rather, our data show that *FOXA1* ablation increases IRF1 levels and subsequent binding to the *CD274* promoter. Interestingly, our results show that this occurs in an interferon-independent manner following *FOXA1* ablation. This data is included in a new Figure 7.

We hope that our responses satisfy the reviews' concerns.

Reviewers' Comments:

Reviewer #2:

Remarks to the Author:

The authors have adequately responded to my queries

Reviewer #3:

Remarks to the Author:

The revised manuscript has a lot of supporting data and supports the conclusions being made. I have no further concerns.